# Joint Tissues: Convergence and Divergence of the Pathogenetic Mechanisms of Rheumatoid Arthritis and Osteoarthritis

**DOI:** 10.3390/ijms26178742

**Published:** 2025-09-08

**Authors:** Marina O. Korovina, Anna R. Valeeva, Ildar F. Akhtyamov, Wesley Brooks, Yves Renaudineau, Gayane Manukyan, Marina I. Arleevskaya

**Affiliations:** 1Central Research Laboratory, Kazan State Medical Academy, Kazan 420012, Russia; koporulina.mo@gmail.com (M.O.K.); anna-valeeva@mail.ru (A.R.V.); 2Institute of Fundamental Medicine and Biology, Kazan (Volga Region) Federal University, Kazan 420008, Russia; 3Department of Applied Ecology, Institute of Environmental Sciences, Kazan (Volga Region) Federal University, Kazan 420008, Russia; 4Department of Traumatology, Orthopedics and Surgery of Extreme Conditions, Kazan State Medical University, Kazan 420012, Russia; yalta60@mail.ru; 5Department of Chemistry, University of South Florida, Tampa, FL 33620, USA; wesleybrooks@usf.edu; 6Department of Immunology, School of Medicine, CHU Toulouse, INSERM U1291, CNRS U5051, University Toulouse III, 31000 Toulouse, France; renaudineau.y@chu-toulouse.fr; 7Laboratory of Molecular and Cellular Immunology, Institute of Molecular Biology NAS RA, Yerevan 0014, Armenia; gaya.manukyan@gmail.com

**Keywords:** osteoarthritis, rheumatoid arthritis, gene mutations, trigger factors, signaling pathways, joint tissues

## Abstract

Rheumatoid arthritis (RA) and osteoarthritis (OA) are frequently occurring multifactorial diseases affecting joints. OA and RA may share not only tissue locations but also some molecular mechanisms. We compared different pathologies: anti-cyclic citrullinated peptide antibody (ACCP)-positive RA—the classical ‘antigen-driven’ pathology, starting in synovia with no signs of inflammatory process; ACCP-negative RA, starting with synovial inflammation triggered by nonspecific factors, which becomes a chronic process due to inherited innate immune peculiarities; and OA, starting with inadequate chondrocyte functioning and cartilage degradation with inflammation as a driving force. Notable coincidences in RA and OA development were revealed: shared mutations of 29 genes encoding molecules involved in immune-inflammatory processes and in ECM production; unidirectional association of OA and ACCP-negative RA with non-genetic triggers; and overactivation of signaling pathways with the same consequences for RA and OA. Innate and adaptive immune responses were involved in OA development. Similar to that observed in RA, lymphoid nodular aggregates were revealed in 30% of OA synovia. Myeloid, and especially pauci-immune and fibroid synovial pathotypes, are possible in OA. Indistinguishable from that in RA, pannuses were found in OA articular tissues. Thus, these coincidences may be evidence of evolution of some OA variants in RA.

## 1. Introduction

Rheumatoid arthritis (RA) and osteoarthritis (OA) are two widespread multifactorial diseases that affect joints. In recent years, the perception of OA as a purely degenerative process has changed. The inflammatory process in the arthritic joint is recognized as the more important component of the pathogenesis and even a trigger for OA [1]. The well-known players in RA pathogenesis and the targets for biological disease modifying antirheumatic drug, TNFα and IL-1β, were recognized as drivers of catabolic signaling in OA joints [2,3]. On the other hand, RA is one of the causes of secondary OA, and the RA inflammatory process can be the driving force of such OA development. In spite of the well documented dependence of secondary OA symptoms on RA duration [4], changed serum levels of various cartilage turnover markers (N-terminal propeptide of collagen IIA, cross-linked C-telopeptide of collagen II, oligomeric matrix protein) in correlation with radiographic/MRI joint damage signs, signifying possible OA initiation, were demonstrated to occur early in RA patients [5,6]. Curiously, as many as 20% of 247 persons at the 3rd (arthalgia) and 4th (undifferentiated arthritis) preclinical RA stages exhibited joint symptoms that were provoked by unusual excessive joint activity, while among 461 eRA patients, there were none in which such activity was identified as a trigger of the disease [7]. The question arises whether imbalance appears in the pre-RA stage and whether it is one of the triggers of RA.

The recognition of inflammation as an OA driver has brought about the idea to apply the groups of drugs used in RA to OA therapy—corticosteroids, methotrexate, biologics [8,9,10,11,12,13]. The results of primarily short-term therapy (4 weeks–6 months) presented in the references mentioned above and a number of other publications are contradictory. Therefore, there is a need for long-term trials.

An assumption can be made that contradictory results are due to differences in the patients included in the study. The difficulty is that OA as a multifactorial disease might have various phenotypes. The existence of different RA phenotypes is not in doubt. The two most well characterized phenotypes are the variants with or without antibodies against cyclic citrullinated peptides (ACCPs) with the obvious differences of the initial pathogenic mechanisms discussed in this review.

Ideas regarding different variants of OA are still in the hypothesis development stage. The following options may be suggested: post-traumatic OA with probable disturbances in repair processes after joint injury; mechanical variants due to the chronic microtrauma of the joint, aggravated by excess weight and other environmental and individual factors [14,15]; and primary hand erosive osteoarthritis with a probable major contribution of immunological mechanisms in its pathogenesis (in particular, the frequent presence of lymphoid follicles in joint tissues, which brings it closer to RA) [16].

So, even a cursory glance allows one to see a certain similarity between OA and RA. These diseases may share not only the arena in which they unfold but also some pathogenic mechanisms. Therefore, it appears reasonable to consider the interweaving of OA and RA pathological mechanisms, which, to some extent, may lead to the convergence of their clinical features.

Accordingly, we will consider the similarities and differences of (i) genetic factors predisposing to the development of OA and RA, (ii) individual and environmental triggers of diseases, as well as (iii) the processes of disease development in joint tissues.

## 2. Comparison of RA and OA Genetic Backgrounds

Genome-wide association studies (GWASs) have provided significant insights into the shared genetic underpinnings of RA and OA. Several single-nucleotide polymorphisms (SNPs) have been identified that are common to both diseases, suggesting overlapping genetic factors that influence susceptibility to joint diseases (Figure 1). Within the 1249 genes identified, 29 genes were found to be common to both pathologies (Figure 1a).

A number of these genes—type II collagen *(COL2A1*), type XI collagen *(COL11A2*), and CUB and Sushi Multiple Domains 1 (*CSMD1*)—play crucial roles in cartilage integrity and extracellular matrix formation. Type II collagen is the main extracellular matrix (ECM protein), while the inclusion of a minor type, XI collagen, is due to some physical properties of mature ECM [17]. Interestingly, XI collagen is used in a rat model for the induction of chronic arthritis [18]. CSMD1 is, in particular, a transmembrane inhibitor of the classical and lectin complement pathways [19] involved in modulating cartilage immune environment due to aggravating of its degradation [20]. SMAD family member 3 (*SMAD3*) and transforming growth factor alpha (*TGFA*)-coded molecules are the factors of TGF signaling pathways, which, being overactivated both in OA and RA, are involved in ECM neogenesis as well as in other pathological processes (neoangiogenesis, apoptosis, and osteoblast differentiation; Figure 2).

Additionally, anaphase-promoting complex subunit 4 (ANAPC4), SUMO-specific peptidase 1 (SENP1), DLG-associated protein 2 (DLGAP2), BLK proto-oncogene, Src family tyrosine kinase (BLK), and *PTPRM*-coded factors are implicated in the dysregulation of apoptotic pathways, disrupting cell cycle regulation in synovial cells, causing increased apoptosis in chondrocytes contributing to cartilage loss in OA and disturbed cell cycling in synovial fibroblasts, and promoting hyperplasia in RA (Appendix A). The *ANXA3*-coded molecule collaborates with RANK in acceleration of osteoblast differentiation [21] and contributes to cell proliferation and angiogenesis via the HIF-VEGF signaling pathway [22]. TYK2 as well as CSMD1 code factors modulating the JAK/STAT cascade (KEGG-obtained data) [20,23], overactivated in both OA and RA *DLG2* and *ITPR3* gene-coding factors, interfering with calcium-dependent signaling pathways regulating ion channel functioning (PRR signaling and inflammasome formation as examples) [24]. FTO Alpha-ketoglutarate-dependent dioxygenase (FTO) is involved in repair of alkylation lesions in DNA, RNA, and nucleoprotein complexes, regulating adipogenic pathways (in particular in pre-adipocyte differentiation) and inflammation-associated vascular endothelial dysfunction [25,26,27]. Besides the more thoroughly studied function of FTO/IRX3 is the involvement in adipocyte precursor development and obesity and diabetes pathogenesis [28,29,30]. A meta-analysis revealed association of *FTO* gene SNPs with hip/knee OA, collagen formation, and extracellular matrix organization biological pathways [31,32]. In addition, it was demonstrated that FTO overexpression alleviates OA progression [33,34]. In RA, FTO SNPs were associated with joint damage, due to the inflammation activity [35]. Increased FTO expression in RA synovial cells enhanced their proliferation and migration and decreased senescence and apoptosis [36].

Two chemokine gene SNPs—C-C motif chemokine receptor 3 (*CCR3*) and C-C motif chemokine receptor 2 (*CCR2*)—may be involved due to the attraction of leucocytes in inflammatory joint loci both in OA and RA, and together with SNPs of the *IL4* gene and its receptor, as discussed above, a SNP of *TYK2*-coding protein associated with the cytoplasmic domain of type I and type II cytokine receptors, and being a part of IFN type I and type III signaling pathways (NCBI), demonstrates the involvement of the immune system not only in RA but in OA as well. Additional notable coincidences are gene mutations that may be important for the development of the adoptive immune response both in OA and RA genes of molecules involved in B-cell development and B-cell receptor signaling—phospholipase C like 2 (*PLCL2*), BLK proto-oncogene, Src family tyrosine kinase (*BLK*) and the two genes located in MHC, class II–butyrophilin like 2 (*BTNL2*), encoding type I transmembrane protein involved in immune surveillance, serving as a negative T-cell regulator by decreasing T-cell proliferation and cytokine release, and *HLA-DPB1* (major histocompatibility complex, class II, DP beta 1) presenting peptides derived from extracellular proteins by B lymphocytes, dendritic cells, and macrophages (NCBI) [31,37,38,39,40,41,42]. The above-mentioned CSMD1 factor promotes both B-lymphocyte receptor signaling and pathways related to antigen presentation [20].

Involved in both pathologies, long non-coding RNAs—LINC02341 and LINC01016 (lncRNAs)—are involved in regulation of processes of cell survival, apoptosis, and lipid metabolism [43,44,45].

STRING analyses (Search Tool for the Retrieval of Interacting Genes/Proteins) of genes showed a distinct clustering of genes involved in both RA and OA with associated SNPs. STRING identifies functional associations between gene products and predicted protein–protein interactions. The analyses visualize (Figure 1b) complex interaction networks by integrating data taken from GWASs. STRING integrates proteins encoded by genes associated with SNPs (Appendix A) to infer how genetic variations might affect protein interactions and pathways [46]. The biggest clustering was obtained for 13 gene products (HLA-DPB1, BTNL2, TSBPI, COL2A1, COL11A2, SMAD3, TYK2, TGFA, CCR2, CCR3, IL-4, IL-4R, and BLK) having protein–protein interactions and associated with biological processes including Th17 cell differentiation and the regulation of leukocyte degranulation. The presence of gene products such as HLA-DPB1, BTNL2, TYK2, CCR2, CCR3, and IL-4R supports immune inflammatory mechanisms that are central to both RA and OA. Numerous studies have emphasized the important role of Th17 cells in driving autoimmune responses in animal models of autoimmune arthritis. In humans, it was shown that Th17 cells correlate with the disease activity [47]. Despite this, recent studies have reported the presence of Th17 cells in the synovial fluid and synovial membranes of OA patients, although in smaller quantities compared to those found in RA joints [48]. Notably, HLA-DPB1 and BTNL2 are involved in antigen presentation and T-cell activation, further underscoring the role of immune responses in joint disease progression. Gene products such COL2A1 (type II collagen, a primary collagen found in cartilage) and COL11A2 (type XI collagen, a minor component of cartilage collagen, working with COL2A1 to maintain ECM stability) play crucial roles in cartilage integrity and extracellular matrix formation. Both RA and OA, despite their distinct pathophysiologies, are significantly implicated in cartilage damage, and the alterations in these proteins across RA and OA indicate a convergence of molecular pathways, particularly those involved cartilage degradation. SMAD3 and TGFA regulate ECM production and repair, contributing to cartilage repair dysfunction in OA and fibrosis in RA.

The second cluster, containing FTO, GLIS3, and JAZF1, is a smaller cluster but highly relevant to the regulation of metabolic processes and cellular function. Particularly, FTO is associated with body mass index (BMI) and obesity, which are known risk factors for OA and potentially for RA. GLIS3 and JAZF1 are associated with gene regulation and metabolic processes, suggesting that metabolic dysregulation may contribute to the development of both RA and OA, albeit through different mechanisms.

The third cluster comprises two gene products, DLGAP2 and DLG2. While the exact roles of DLGAP1 and DLG2 are involved in regulation of apoptosis and calcium-depended signaling cascades, including inflammasome formation—the processes undoubtedly being important links in the pathogenesis of both RA and OA.

The clustering analysis suggests that the shared genetic factors and their products could provide new insights into the pathogenesis of these diseases. Further research is needed to validate these findings and determine how these shared genes contribute to disease onset and progression in RA and OA.

## 3. Signaling Pathways in RA and OA Pathogenesis

The following signaling pathways are considered fundamental in RA and OA pathogenesis: NF-κB, Wnt, Jak/STAT/SOCS, OPG/RANKL/RANK, TGFβ/ΒΜP, and HIFs-PHDs [49,50,51] (Figure 2, Figure 3 and Figure 4).

As shown in Figure 2, created on the basis of the schematics presented in Wikipedia with the addition of KEGG information, these signaling pathways are closely intertwined and interact in the process of modulating precisely those mechanisms, which, when disrupted, play a role in the pathogenesis of both OA and RA. We attempted to analyze the participation in OA and RA pathogenesis from the point of view of the progression of destructive processes in OA and RA joints.

Osteoarthritis. Excessive NF-κB signaling in synovial tissue samples has been detected already in the early stages of OA, and it is probably not limited to the synovium, but extends at least to cartilage and subchondral bone with chondrocyte macrophages, synoviocytes, and osteoblasts in a vicious circle of mutual potentiation of proinflammatory activities [51,52,53,54,55,56,57]. Overactivation of NFkB and of interdependent signaling pathways is triggered by abnormal mechanical pressure and hypoxia and interaction of crystals, cartilage fibronectin, hyaluronan, biglycan, tenascin c, syndecan-4, and type II collagen and aggrecan fragments with PPRs [51,58,59]. The abnormal excessive signaling in OA might be due to TLR9 and TLR3 SNPs, associated with OA risk [60,61], numerous gene mutations of the factors of TGF cascade associated with cartilage thickness, erosive hand OA or hip minimal joint space width (PIK3R1, SMAD3, CEMIP (KIAA1199) BMP6, and NOG) partly presented in Appendix A [62,63,64,65] and probably HIF SNP (contradictory data) associated with OA and RA disease risk [66,67].

Excessively activated NFkB cascade acting directly or in collaboration with overstimulated JAK/STAT, RANKL/RANK, a branch of HIFs-PHDs (HIF-2α) and WNT, dysregulated TGFβ/ΒΜP is due to increased production of matrix metalloproteinases (MMP1, MMP2, MMP3, MMP7, MMP8, MMP9, MMP13), ADAMTS4 and ADAMTS5, degrading articular cartilage. Various inflammatory molecules (prostaglandin E2, nitric oxide synthase and nitric oxide, cyclooxygenase-2), induced by hyperactivated NFkB cascade promote chondrocyte apoptosis in the inflammatory focus. The abnormal signaling in a tangle of RANKL/RANK, HIFs-PHDs, JAK/STAT, TGFβ/ΒΜP, and WNT pathways results in chondrocyte dedifferentiation and loss of functions—decreased expression of COLII, aggrecan, Sox-9, collagen I collagen, and X collagen [50,51,68,69,70,71,72]. Abnormal RANKL/RANK and WNT signaling in OA proinflammatory osteoblasts is due to subchondral bone loss and bone sclerosis development [51]. Another subchondral event—osteophyte formation—occurs with chondrocyte TGF cascade participation [73,74]. Neoangiogenesis in subchondral bone with vessel penetration into normally avascular cartilage and synovia is due to proinflammatory signaling in the discussed cascades with the HIF pathway first of all. Proinflammatory cytokines stimulate synovial macrophages, abnormal OA osteoblasts, and chondrocytes to produce vascular endothelial growth factor (VEGF)—a major actor of neovascularization [75,76].

Rheumatoid arthritis. The accumulation of gene polymorphisms of factors of the discussed signaling pathways, with the possible exception of the HIF cascade, is noteworthy [40,77,78,79,80,81,82,83,84,85,86]. OA-associated SNPs in signaling pathways have been found significantly less frequently, or this aspect of OA pathogenesis has been less studied. In addition, RA is characterized by a number of mutations in TLR genes [87]. Signaling from overexpressed PRRs, in particular in fibroblast-like synoviocytes, macrophages, and other joint tissues [87], in combination with SNP signaling pathway factors, lead to a rampant proinflammatory reaction in the joints. It is probably more pronounced than in the OA activation of the same tangle of signaling pathways and leads to practically the same consequences—increased production of matrix metalloproteinases and ADAMTs, chondrocyte degradation/survival, ECM degradation, osteoclastogenesis, subchondral bone loss and bone sclerosis development, neoangiogenesis, fibroblast proliferation, and fibrosis [84,88,89,90,91,92,93,94,95,96,97,98,99,100,101,102,103,104].

Thus, there is a clear similarity between the role of overactivation of signaling pathways and the consequences of this activation for joint tissues in RA and OA. The signaling pathways being RA hallmark and the application points of its modern biologic and targeted therapy are the drivers OA as well. Overactivation of these signaling pathways is associated with the involvement of inflammation and immune processes in the pathogenesis of both diseases (indications of immune system involvement in the pathogenesis of OA are discussed below). Yet there is an obvious difference. RA is an immune inflammatory disease characterized by a powerful systemic inflammatory process [105,106]. OA is characterized by low-grade inflammation, which is not always determined outside the joint tissues and has less pronounced signs of innate and cellular immune mechanism involvement [107].

## 4. Individual and Environmental RA and OA Triggers

As reported in other multifactorial diseases, both RA and OA are provoked by interaction of external factors with the unique genetic features of the diseases. As it was mentioned above, there are two well-identified RA variants—ACCP-positive and ACCP-negative—with the discussion below focusing on some essential differences of their genetic backgrounds and early-stage pathogenic mechanisms, while OA variants are currently discussed mainly at the level of analyzing clinical data and developing therapy without depending on the differences in the intimate pathogenic mechanisms, which are now beginning to be intensively studied [108,109,110].

Therefore, we tried to compare the influence of the main non-genetic triggers on the development of OA and two variants of RA—ACCP-positive and -negative—using PubMed as a source of information. The following key words were selected: “disease, trigger, statistical indicator” (Appendix A).

It should be noted that the different contributions of non-genetic factors to the development of OA and ACCP-positive and -negative RA variants primarily can only be assumed based on the noticeable differences in OR/HR/RR values, as we failed to find any publications with a rigorous analysis of this hypothesis (Appendix A).

The risks of both RA variants in the presence of these diseases in the family history were increased [111,112,113]. The authors did not conduct a direct comparison of the family risk in the ACCP-positive and ACCP-negative RA cohorts depending on the age of onset of the disease; however, upon detailed consideration of the presented data, the following hypotheses arise. First, the risk of developing RA in the presence of a relative with ACCP-positive RA in the family is higher than in the presence of ACCP-negative cases in the family history [111]. Second, in both cohorts, the risk of developing RA is the highest if the disease develops in the proband at an age of less than 40 years.

If there are cases of OA in the family, the risk of developing this disease in relatives is also significant [112,113].

Risk of both ACCP-positive and -negative RA onset increases with age, yet one could notice some difference. The risk of ACCP-positive cases was the highest at the age of 50–59 years and then appears to decrease, while in the ACCP-negative cohort, it was high at the age of 45 years and increased with age until at least 60–64 years [114]. The incidence of OA also increased steadily with age [115].

Women are more prone than men to develop RA and/or OA [113,116,117,118,119]. Yet, only a modest role of gender differences in OA and even the absence of a reliable difference in the frequency of hand OA, in contrast to hip and knee involvement, were revealed [118,119]. Perhaps this is one of the arguments for the existence of different variants of OA with specific triggers, mechanisms of progression and, possibly, requiring different therapeutic approaches.

Gender preferences and the link of both RA and OA with the corresponding age ranges evoke a natural assumption that the changes in sex hormone levels in pre- and perimenopause and later on may provoke the diseases. However a trigger role of this factor was demonstrated for OA and ACCP-negative RA, but not for the ACCP-positive one [114,120,121]. The results of the analysis of the postmenopausal replacement hormone therapy link with ACCP-negative and -positive RA and OA cohorts were contradictory, which may be due to the differences in pill compositions, doses, duration of the therapy, and, as OA, localization of the process presented in the analysis [114,120,122,123,124,125,126].

The triggering role of parous in RA risk may depend on the number of children as well as on the normal/adverse pregnancy data [127]. Yet the analysis revealed that being parous increased ACCP-negative RA risk with no impact on ACCP-positive one [128]. In addition, being parous was a predictor of a more severe RA among ACCP-negative younger women [129]. OA risk also increased in parous vs. non-parous women and depended on the number of children [130,131]. Breastfeeding association with RA depended on its duration and may have a protective effect for ACCP-positive cases, with no impact on ACCP-negative cases [132,133,134,135]. Few publications demonstrated a provoking effect of breastfeeding on OA risk [130,136]. The analysis of oral contraceptive usage impact on ACCP-positive RA risk yielded contradictory results and revealed no effect on ACCP-negative RA and OA [124,131,132,133].

Smoking by the parent during pregnancy (in utero exposure) increased the risk of ACCP-positive RA with no impact on the risk of ACCP-negative cases (no data were available regarding the link with OA risk) [137]. Larger gestational age birth weight increased RA risk regardless of ACCP status, while smaller gestational age birth weight appeared to be protective [135,138,139]. The opposite pattern was found for OA [140,141,142].

The curious patterns of BMI links with the risks of OA and RA variants were revealed. Being overweight had no impact on the risk of ACCP-positive RA cases or even had a protective effect [117,132,143], while it was associated with the risk of ACCP-negative RA cases [132,143,144]. Risk of knee and hip OA, as expected, increased with overweight (conflicting data regarding hip OA), due to the overload of the lower limb joints [113,145,146,147,148] However, being overweight has also been associated with an increased risk of hand OA, which clearly suggests a more complex relationship than excess joint loading [148,149].

Coffee (more than 10 cups per day) in a dose-dependent pattern increased the risk of ACCP-positive cases and had no effect in ACCP-negative cases [132]. Only marginally significant evidence of the link of low daily coffee doses and OA risk and not in all studies was revealed [150,151]. Consumption of ≥7 cup of coffee was linked with increased OA link in men, but not in women [151].

Alcohol consumption dose-dependently reduced the risk of developing ACCP-positive RA, but was associated with an increased risk of developing its ACCP-negative variants in patients with >15 drinks per week [117,144,152] and was a risk factor for severe erosive hand OA [153,154].

Smoking was definitely associated with an increased risk of ACCP-positive RA especially in shared epitope (SE) carriers [132,155,156]. Analyses revealed a possible protective effect of smoking on the risk of ACCP-negative cases without SE in the genome and no link with that in SE carriers [132,155,156]. A possible protective effect of smoking was demonstrated for knee OA as well [157,158,159].

Mental stress prior to RA onset was definitely a trigger, and antidepressant use had a protective effect [160,161]. However, upon further examination, it was found that mental stress was associated with the development of ACCP-negative RA, but not the ACCP-positive one [162]. Mental stress was associated with the severity of radiographic features of knee/hip OA, but the study design did not allow us to understand whether mental stress was a risk factor for OA progression, or whether symptoms due to progressive joint disease were the cause of stress [163,164].

Ten years of high and even slight physical activity wer a risk factor for ACCP-negative but not for ACCP-positive RA [132]. And the various variants of work and everyday physical activity were a major risk factor for OA [113,165,166].

So, two RA variants and OA are predominantly “female” family-aggregated age-related diseases. Yet their links with non-genetic triggers appeared to have certain differences, probably reflecting the difference in the contribution of concrete pathogenetic mechanisms to their development.

Some patterns emerge when comparing the lists of RA and OA triggers. Family clustering has a complex nature, including shared genetics; infections; microbiome; lifestyle; and ecology [167]. Familial aggregation of RA and OA may be due to different triggers from this list.

The peak risk of developing both RA and OA occurs in the age range of 50–60 years. The well-known molecular mechanisms of aging are the same for OA and RA with a greater or lesser contribution to their development. Imbalance in production and inactivation of reactive oxygen species due to chondrocyte cell death and ECM degradation [168,169]. Oxidative stress is due to DNA, lipid, and protein damage, leading to synovial inflammation, essential for the both diseases [170]. Reactive oxygen species are involved in the process of carbamylation, which provokes the production of antibodies to carbamylated peptides, which have diagnostic and prognostic significance in RA [171,172]. Oxidative stress might trigger joint symptoms in patients at preclinic RA stages [173]. Immune system aging results in proinflammatory shift of immune reactions [174]. Tissue damage and increased proinflammatory potential of immune system predisposes for chronic-persistent inflammation.

Common sense would expect traditional lifestyle differences and gender-based occupational characteristics certainly contribute to the female/male disparities in RA and OA. Yet, it is noteworthy that the ACCP-negative RA variant appeared to be more dependent on fluctuations in estrogen levels than the ACCP-positive cases. The same trend was observed in OA. Indeed, menopause is characterized by a decrease in estrogen and progesterone levels, and a weakening of their anti-inflammatory effects [167]. Being parous is associated with OA and ACCP-negative RA, but not with ACCP-positive cases. Normal pregnancy is characterized by increased estrogen and progesterone levels and Th1→Th2 immune responses, and with the delivery process, there is a powerful explosion of Th1 responses against the background of decreasing estrogen and progesterone levels. So, repetitive surges of a Th1 response at each delivery might ultimately trigger a persistent immune inflammatory process, characteristic primarily of ACCP-negative RA and, perhaps, to some extent, of OA. Breastfeeding is characterized by high levels of prolactin supporting a Th1 response by reduced levels of estrogen and prolactin compared to the period before pregnancy (which quickly return to normal in the absence of breastfeeding). So, this situation, if repeated, is also fraught with the possibility of provoking a chronic immune-inflammatory response. This is a very simplified speculative scheme of the possible involvement of sex hormone-associated processes in provocation of ACCP-negative RA and OA. Further studies are needed to clarify the possible gender-dependent differences between ACCP-positive and ACCP-negative disease variants.

Obesity and mental stress, which can increase risk for OA and ACCP-negative RA, but not ACCP-positive RA risk, also have proinflammatory potential [167]. Another trigger of OA and ACCP-negative, but not ACCP-positive RA is physical activity, which may impact the diseases via repetitive joint tissue microtrauma and concomitant inflammation. The scheme of the greater importance of non-specific proinflammatory triggers of OA and ACCP-negative, but not ACCP-positive RA, contradicts the links between smoking, which provokes ACCP-positive RA, but possibly has a protective effect in ACCP-negative cases and OA, and alcohol consumption, which reduces the risk of developing ACCP-positive RA and provokes ACCP-negative cases and OA. The provoking effect of smoking in ACCP-positive RA was expected given its well-known association with ACCP production [132,155,156]; however, its other known mechanisms are proinflammatory [167]. The protective effect of alcohol consumption on RA development is usually associated with its anti-inflammatory effect [167]. However, it is not clear why this effect occurs in ACCP-positive cases but not in ACCP-negative RA or OA.

Despite the obvious contradictions, possibly related to the lack of data analyzing the differences in the connections of triggers with ACCP-positive and ACCP-negative RA and OA, a certain trend is visible—the lists of triggers for OA and ACCP-negative OA are similar.

Adipose tissue was recognized to be an important player in OA pathogenesis. In addition to the obvious impact of obesity as a trigger for the development and progression of the disease, there are also less obvious connections. In particular, interesting data were obtained in experiments on mice with lipodystrophy, protected from spontaneous and posttraumatic OA even in the presence of known trigger factors—obesity, systemic inflammation, and high-fat diet [175]. Implantation of adipose tissue or fibroblast-derived adipocytes restored susceptibility of the mice to post-traumatic OA. Adipocytes produce a number of well-studied adipokine modulators of various processes in the body. They interact with sex hormones, interfere with immune processes and inflammation, and interact with chondrocytes (Table 1). These functions make them undoubted participants in the pathogenesis of OA and RA.

In terms of OA and RA pathogenesis, and of the various functions of adipokines, their pro/anti-inflammatory potential and effect on cartilage are of interest. When analyzing the influence on various immune processes, adipokines are very conditionally divided into predominantly proinflammatory (adipsin, leptin, resistin, visfatin, chemerin), with those discussed above obviously promoting both RA and OA progression; anti-inflammatory (apelin, vaspin); and multidirectional (adiponectin, omentin). All the analyzed adipokines with proinflammatory effects as well as adiponectin promoted ECM degradation via various mechanisms. Anti-inflammatory apelin demonstrated a multidirectional yet mainly catabolic effect—stimulated chondrocyte proliferation, yet increased expression of MMP and IL-1beta and decreased collagen II level. Omentin and vaspin contributed to the replenishment of cartilage tissue or, at least, prevented its degradation.

Due to their participation in modulation of inflammation and cartilage processes, adpokines play a significant role in the both RA and OA pathogenesis. Promoting cartilage degradation, blood and synovial fluid proinflammatory adipokines and multidirectional adiponectin were increased and correlated with RA activity and OA progression. Oppositely acting vaspin and omentin were decreased in blood and synovial fluid and inversely correlated with RA activity and OA progression.

Recognized as an anti-inflammatory adipokine, apelin, again, does not fit into these patterns. Although its level was reduced in RA, it promoted neoangiogenesis, being a major component of joint damage. In OA, increased apelin levels were demonstrated due to the joint damage progression via the same mechanism, neoangiogenesis promotion.

Adipokines undoubtedly are the major players in obesity, play role in aging, and interplay with sex hormones, and so, participate in the implementation of the trigger role gender and gender-associated risk factors (Table 1). So, they might be involved in implementation of OA and RA triggers. Proinflammatory adipokine levels were directly associated with weight gain, while the link of adiponectin and omentin with BMI was inverse. Contrary to expectations, levels of anti-inflammatory adipokines also increased with weight increasing, possibly regardless of their impact on the inflammatory process. Despite the ambiguous patterns, adipokines are clearly powerful players in the pathogenesis of obesity and undoubtedly promote the implementation of the role of obesity in OA and RA provocation. Proinflammatory adipokine, adiponectin, and omentin expression was associated with aging or at least age-associated diseases. Anti-inflammatory vaspin demonstrated the same pattern, while apelin was inversely associated with age. So, the age-associated dynamic of proinflammatory adipokines and adiponectin is in good agreement with the role of age in the provocation of RA and OA and the relationship of these factors with the parameters of these diseases. The inverse association of apelin with age, its anti-inflammatory functions, and clear link with RA and OA are also in good agreement with each other. Omentin and vaspin expression in aging and their link with OA and RA parameters, at first glance, appear paradoxical. Perhaps this might be due to the change in tissue sensitivity to these factors (in particular, the age-dependent change in the expression of receptors to them).

Analysis of gender disparities of adipokine levels in connection with sex hormones led to ambiguous results. Leptin, adipsin, chemerin, and adiponectin levels in females were higher than that in males, and these findings agree with their role in OA and RA pathogenesis as well as with the fact that these diseases show a female bias. In addition, adiponectin, adipsin, leptin, and chemerin expressions were in direct connection with estrogen and inversely with testosterone levels. However, gender disparity links with sex hormones were not so straightforward. Omentin demonstrated an inverse correlation with gender dependence, which is consistent with its interference with OA and RA mechanisms, yet its levels were in direct correlation with estrogen. Apelin and visfatin were also in direct connection with estrogen and inversely with testosterone levels, yet no sexual dimorphism of these adipokines was revealed. The same patterns were demonstrated for resistin. On the contrary, vaspin levels were independent from estrogen and progesterone, yet they were higher in females vs. males.

The ambiguous picture of the connections of adipokines with sex differences, contradicting their impact to the parameters of “female” diseases of OA and RA, is likely to be due to the following reasons: (i) gender dependence of these diseases has a comprehensive nature—at least lifestyle, eating habits, and professional differences undoubtedly contribute to their provocation; (ii) as discussed above, sex hormones, regardless of their connection with adipokines, modulate immune reactions; (iii) interplay of adipokines and hormones and their interference in the mechanisms of development of the diseases under discussion might be of different value in OA and RA phenotypes.

Another important factor is that the polymorphisms of genes of some adipokines may affect to susceptibility to rheumatoid arthritis and osteoarthritis. It is worth noting that heterogeneity between the study’s results was observed in the analysis (Table 2).

Discordant results are common among genetic studies on complex diseases. Possible explanations for controversial results include clinical heterogeneity, ethnic differences, real genetic heterogeneity, and small sample sizes. Geographical regions may be the source of heterogeneity in studies of polymorphisms of adipokines and RA/OA susceptibility.

**Table 1 ijms-26-08742-t001:** Adipokines role in RA and OA.

Adipokine	Immune Functions	Cartilage	Estrogens/ Testosterone	Female/ Male	Age	Lean/Obese	RA	OA
Adiponectin	Multidirectional [176]	Promoted aggrecan degradation [177]	⇑/⇓ [178,179,180]	f > m [178,179,180,181]	⇑ 66–80 years vs. 51–65 years = 36–50 years [182]	⇓/⇑ [183] ⇑/⇓ [184,185]	⇑ CRP and ESR [186,187,188]	⇑ cartilage damage [177,189]
Omentin	Multidirectional (⇑ IL-4, ⇑ IL-1β) [190]	Blocks cartilage degradation, bone erosion, chondrocyte senescence via suppressing the proinflammatory cytokines [191,192]	⇑/? [193]	f < m [194]	⇑ [195]	⇑/⇓ [190,196,197]	⇓ MMP-3 levels, RA activity CDAI, ESR [198]	⇓ in synovia [199] ⇓ OA progression [191]
Apelin	Anti-inflammatory [200,201]	In total catabolic: stimulated chondrocyte proliferation, yet increased expression of MMP and IL-1beta and decreased collagen II level [202]	⇑/⇓ apelin and APJ expression are up-regulated by estrogen [203] Inverse association with testosterone levels [204]	No sexual dimorphism [205]	⇓ apelin and its receptor (APJ) expression [206,207]	⇑/⇓ No difference [183] ⇓/⇑ [208,209]	⇓ [210] Promotes neoangio-genesis [211]	⇑ progression via stimulation of neoangio-genesis [212]
Vaspin	Anti-inflammatory [213]	Promoted differentiation and chondrocyte survival, and ECM formation [214]	No association [215]	f > m [216,217]	⇑ [182]	⇓/⇑ [217]	⇓ eRA activity (DAS28), ESR, CRP levels [218]	⇓ in serum ⇑ in synovia [219]
Adipsin	Proinflammatory [190]	Promoted cartilage volume loss [220]	⇑/? expression of adipsin gene [221]	f > m [222]	⇑ [223,224]	⇓/⇑ [223]	⇑ clinical activity in early RA [225]	⇑ +OA progression [220]
Leptin	Proinflammatory [190]	Promoted chondrocyte apoptosis [226], degradation ECM [227], cartilage volume loss [228]	⇑/⇓ [193,229,230]	f > m [180,231,232,233]	⇑ In male ? In female leptin resistance due to reduced expression of leptin receptor [234]	⇓/⇑ [185,190,235]	⇑ [186,187] Direct link with CRP levels [188]	⇑ [193] Prediction of early-onset post-traumatic OA [236]
Resistin	Proinflammatory [190]	Promoted proteoglycan loss due to inhibition of proteoglycan synthesis in chondrocytes [237,238]	⇑/⇓ [215,239]	f < m [240] f = m [241]	Associated with combination of age-related comorbidities but not with age itself [242,243]	⇓/⇑ [190,244]	⇑ [186] Direct link with CRP levels [188]	⇑ [193,245] Prediction of early-onset post-traumatic OA [236]
Visfatin	Proinflammatory [190]	Promoted collagen II and aggrecan degradation [246]	⇑/⇑ [247,248]	f = m [249,250]	⇑ In female [250]	⇓/⇑ [190,251]	⇑ Direct link with DAS28 and CRP [187,252]	⇑ [193] Direct link joint damage [246] Prediction of early-onset post-traumatic OA [236]
Chemerin	Proinflammatory [190]	ECM degradation due to stimulation of pro-catabolic cytokine and metalloproteinase production [253]	⇑/? [193,254]	f > m [254] f < m [255] f = m [256]	⇑ [257]	⇓/⇑ [190,235,258]	⇑ Direct link with DAS28, ESR, CRP [259]	⇑ [193,245] Prediction of early-onset post-traumatic OA [236]

⇑—under the influence of the factor, the adipokine level decreased; ⇓—under the influence of the factor, the adipokine level increased; ?—no data.

**Table 2 ijms-26-08742-t002:** The influence of polymorphisms of some adipokines genes on susceptibility to RA/OA.

	RA	OA
Adiponectin	Polymorphisms rs266729, rs2241766, rs2082940, and rs1063539 in the adiponectin gene—no association with RA. Adiponectin gene rs1063539 locus was possibly associated with anti-CCP in RA female patients [260]. No significant genetic correlation between adiponectin levels and RA [261].	The ADIPOQ gene rs1501299 (+276G/T) polymorphism was not associated with KOA severity or vulnerability [262]. Polymorphisms +45T/G and +276G/T of the ADIPOQ gene might not be responsible for OA susceptibility among Thais [263]. The SNP rs182052 in the ADIPOQ gene may potentially modify individual susceptibility to knee OA in the Chinese population [264]. Associations may exist between ADIPOQ rs2241766 and knee OA in Asians’ DOI [265]. The ADIPOQ gene rs1501299 polymorphism intensifies the risk of knee OA in this Chinese Han population [266].
Omentin	Revealed the association between omentin rs2274907 and RA susceptibility [267].	The Val109Asp polymorphism of the omentin-1 gene may not be the primary pathogenic factor of KOA in Chinese individuals. The Val/Val genotype can be regarded as a potential biomarker for the risk of KOA progression [268]. ITLN1 (intelectin-1, also known as omentin) polymorphism rs2274908 was related to KOA risk in the Han population [269].
Leptin	Leptin gene (rs10244329, rs2071045, and rs2167270) polymorphisms are not associated with RA genetic susceptibility and its clinical features in the Chinese population [270].	In normal weight and overweight Han Chinese individuals, LEP polymorphisms (three SNPs of leptin—rs11761556, rs12706832, rs2071045) were associated with knee OA [271].
Resistin	There were no significant differences for the distribution of allele and genotype frequencies of three resistin SNPs (rs1862513, rs3745368, and rs3745367) between RA patients and normal controls (all *p* > 0.05). The genotype effects of dominant and recessive models were also analyzed, and no significant association was detected (all *p* > 0.05). Haplotype analysis suggested that the frequency of haplotype GAA was notably lower in RA patients in comparison with normal controls. Thus, resistin gene polymorphisms might affect the genetic predisposition of RA in the Chinese population [272]. C allele of the resistin SNP rs7408174 as well as those with the AG allele or who had at least one A allele of the SNP rs3219175 are at greater risk of developing RA disease compared with wild-type carriers [273].	Weak associations between resistin genes and hand OA in Finnish women, and that the associations are modified by BMI [274]. Resistin −420/+299 alleles haplotype analysis demonstrated that mutant alleles were more prevalent in knee OA-affected individuals compared to healthy subjects (*p* < 0.05) in Pakistani population [275]. SNP rs3745368 from resistin was identified as being related to an increased risk of HOA [276].
Visfatin	X	SNP rs4730153 was significantly associated with decreased risk of OA in an additive genetic model (*p* < 0.05), while rs16872158 showed an increased risk of developing OA (*p* < 0.05) in the Chinese population [277]. Limited data revealed that associations may exist between visfatin rs4730153 and knee OA in Asians, and between visfatin rs16872158 and knee OA in Asians [265].
Chemerin	Chemerin rs17173608 polymorphism were associated with increased susceptibility to RA [267].	X
Apelin	No association between apelin rs2235306 and RA [267].	X

X—no data.

## 5. Comparison of Pathogenesis of ACCP-Positive and ACCP-Negative RA and OA Through the Prism of Joint Tissue Processes

### 5.1. ACCP-Positive RA: Initial Processes

ACCP-positive RA is a classical ‘antigen-driven’ pathology [278]. The cornerstone of its development is cyclic citrullinated peptide (CCP) interaction with the immune system. Initially, CCP portions appear in inflamed oral, lung, and presumably gut mucosa at preclinical stages [279,280]. Presentation of these neo-epitopes by antigen-presenting cells in secondary lymphoid organs, being a common non-specific event, becomes critical in persons at risk of ACCP-positive RA due to a well-known interplay of CCP with T-cell receptors in SE carriers, supported by a vast number of gene SNPs, involved in the shaping selection, maturation, and functioning of T and B cells [281]. Perinatal epigenetic events might also play a role. At least in utero nicotine exposure increased the risk of ACCP-positive RA but not in ACCP-negative cases [137].

The next step in RA promotion is associated with ACCP and specific lymphocyte clones. Studies at the preclinical stage demonstrated that specific CD3+ T lymphocytes with highly proliferative, tissue-invasive, and proinflammatory effector cells non-stochastically migrate into synovia, instead of leading to relatively quiescent memory T cells; ACCP stochastic migration also takes place [282]. A subsequent trigger might lead to synovia infiltration by oligoclonal B cells, together with macrophages, developing a local germinal center (GC), and accumulation of other B-lymphocyte subsets, attracted to synovia by unknown mechanisms [283,284].

Notably, at the preclinical stage, ACCP and lymphocyte oligoclones are found in synovia with no signs of inflammatory processes and vascular bed foreign cell infiltration [282]. These processes develop as a consequence of the ACCP and lymphocyte impact on joint tissues. ACCP deposited in joint tissues may activate osteoclast precursors expressing citrullinating enzymes and surface CCPs [285], due to bone loss and joint pain [286,287,288]. Infiltrating oligoclonal T and B lymphocytes produced proinflammatory factors that become drivers of synovial inflammation [289].

### 5.2. ACCP-Negative RA: Initial Processes

According to Pratt’s model [278], the key node of ACCP-negative RA pathogenesis is dysregulation of CD4+lymphocytes due to SNPs in a number of genes mainly *ANKRD55* and *STAT4*, genes of components of the IL6 signaling pathway strongly associated specifically with this phenotype. Due to the proinflammatory skewing of immune reactions, any non-specific events provoke local inflammation. Unlike ACCP-positive RA, in ACCP-negative cases, a wide range of non-specific cytokine-activated T lymphocytes of various antigen specificity attracted to synovia was demonstrated [290]. In addition to wide T-cell clone numbers, another remarkable peculiarity of synovial processes in ACCP-negative RA is an increased number of dysfunctional cells with lower cytotoxic gene expression and increased expression of exhausted genes [291], and lower antigen processing and presentation activity in synovial B cells than that in the ACCP-positive variant [292,293,294].

Proinflammatory cytokines, predominantly of macrophage origin, with differences in biologic efficiency suggest a greater proportion of myeloid pauci-immune patterns in ACCP-negative synovia vs. lympho-myeloid pattern in ACCP-positive cases [295].

Early-stage ACCP-negative RA demonstrates a greater contribution of innate immune mechanisms [295,296], namely pathogen-associated molecular patterns (PAMPs), TLRs, Nod-like receptors, inflammasomes, complement system, monocytes/macrophages, granulocytes, natural killers, and dendritic cells [297].

Of fundamental importance is that, unlike ACCP-positive RA, ACCP-negative cases start as an immune response to a synovial injury with subsequent inflammation, due to exposure to non-specific factors (trauma, infection) [295].

### 5.3. OA: Initial Processes

Initial events are cartilage damage, decrease in chondrocyte functional activity, and a shift from anabolic to catabolic processes in cartilage tissue due to external influences, namely joint injury or joint tissue microtrauma with repeated loads or overweight, hormonal imbalance, accidental entry fragments, and products of commensal bacteria translocated from mucous membranes to joint tissues with inflammatory focus formation or other triggers of aseptic inflammation [298,299,300,301]. Gene SNPs of the factors involved in ECM formation [15,302] may aggravate these events.

Inflammation may be a trigger factor for functional asthenia of chondrocytes, or it may conjoin later due to proinflammatory factors produced by these cells in unfavorable conditions [303]. At the early stage with minimal radiographic signs of cartilage damage, synovia is involved with signs of ongoing inflammatory and fibrotic processes and perhaps autoimmune reactions—mononuclear infiltrates, diffuse fibrosis, lining layer thickening, macrophage appearance, and neoangiogenesis [304,305,306]. Intensive exchange of proinflammatory signals between degrading chondrocytes and activated synovial cells creates a vicious cycle. Crosstalk between chondrocytes, synoviocytes, osteocytes, osteoblasts, osteoclasts, and endotheliocytes is due to early-stage subchondral bone loss and late-stage bone sclerosis, subchondral bone cysts, bone marrow oedema-like lesions, and osteophyte formation [307,308].

So, despite the obvious dissimilarities of the starting points, chondrocytes and cartilage in OA and synovia in RA, there are some similarities in initial processes in OA and ACCP-negative RA.

## 6. ACCP-Positive and ACCP-Negative RA in Full Swing

Regardless of the characteristics of the initial stages of synovitis development in ACCP-positive and ACCP-negative variants of RA, further events develop in many respects in the same way. The process includes recruitment from the vascular bed and local M1 macrophages and synovial fibroblasts undergoing de-differentiation in response to activating stimuli, mainly originating from tissue-invasive T cells undergoing pyroptosis [309]. Other major players of synovitis development and perpetuation are recruited by cytokines, blood granulocytes, and macrophage. Granulocytes form neutrophil extracellular traps (NETs) [310,311,312], and together with synovial macrophages, endotheliocytes, and fibroblasts, release a bulk of toxic proinflammatory agonists into the surroundings, synergistically destroying synovial cells, cartilage, and bone [313,314,315,316].

The well-known feature of RA synovia is the presence of ectopic lymphoid structures (germinal centers), resembling secondary lymphoid organs with plasma cells, macrophages, B- and T-lymphocyte compartments, follicular T helpers, lymphatic vessels, and high endothelial venules [317,318]. It is important to note that these tertiary lymphoid tissues are not specific for RA, but develop in non-lymphoid tissues in response to chronic inflammation. The cells of lymphoid aggregates produce cytokines and adhesion molecules locally in the joint tissues with a subsequent increase in peripheral blood [318,319,320]. Germinal center cells might produce autoantibodies. The ongoing somatic hypermutation and class-switch recombination of Ig genes and ACCP production by germinal center lymphocytes has been demonstrated [321,322]. B cells differentiated within synovial ectopic lymphoid structures target deiminated proteins that could be generated during NETosis [323,324]. RF production in synovial germinal centers was also assumed [325]. However, these results are contradictory with the fact that the presence of lymphoid aggregates is associated with the inflammation intensity, but not with the ACCP-positive RA phenotype [326,327,328].

It should be specified that lymphoid neogenesis was demonstrated in RA in no more than 30–50% of synovial samples and the presence of follicular dendritic cells as an attribute of a germinal center in less than 10% of RA biopsies [318,329].

RA synovia has some features of a local neoplastic process, such as significantly expressed angiogenesis and aggressive expansion of dedifferentiated fibroblasts with abnormal overexpression of embryonic genes (wnt5A as an example [330]). The leading role in pannus formation is played by fibroblasts with the participation of synovial macrophages. Fibroblasts are effectors of cartilage and bone destruction due to unique invasive abilities, the production of large numbers of matrix-degrading enzymes, and resistance to apoptosis. RA is characterized by an increase in the amount of antiapoptotic proteins, Bcl-2 and Mcl-1, SUMO1, FLIP, as well as by somatic mutation of the p53 protein gene, which probably contributes to the inhibition of synoviocyte apoptosis [331,332,333].

Another well-known feature of both ACCP-positive and ACCP-negative RA variants is development of secondary OA due to the negative impact of inflammation on chondrocytes, osteoblasts, and fibroblasts [305,334,335,336]. OA incidence in a Russian cohort of 620 RA patients was 70%, whereas in the non-RA population, it was 21% [4,337]. The main risk factors for OA progression in RA patients were high RA activity (DAS28), RA experience for more than 10 years, age more of than 45 years, BMI of more than 25 kg/m^2^, and glucocorticosteroid intra-articular injection 6 months or more before the study [4].

Despite the convergence of pathoimmunological mechanisms, some differences are revealed in ACCP-positive and ACCP-negative RA advanced stages. Synovial tissue from ACCP-positive patients had a higher number of infiltrating lymphocytes, less extensive fibrosis, and a thinner synovial lining layer compared with synovial tissue from ACCP-negative patients [338]. The samples of ACCP-negative patients showed an increased extent of fibrosis and a thicker synovial lining layer. These specific features of ACCP-positive and -negative RA synovitis were the same in the 31 synovial samples of the same patients obtained 3–4 years earlier (including those obtained at an early disease stage). The authors also noted that samples of ACCP-positive patients were more likely to have germinal centers, although there were no significant differences in this indicator, probably due to the small number of samples. It should be specified that attempts to link the presence of lymphocytic infiltrates with ACCP-positive RA yielded conflicting results. On the one hand, it was demonstrated that synovial tissues with lymphocyte aggregates contained significantly elevated RF-IgM and anti-CCP IgG compared to tissues with diffuse lymphoid infiltration [322]. On the other hand, it was revealed that lymphoid neogenesis was linked to a higher RA activity but not with the ACCP or RF phenotype [318]. ACCP-positive RA is likely linked with the more pronounced secondary OA progression, assessed using the Kellgren/Lawrence scale or serum levels of cartilage oligomeric matrix protein and other markers of cartilage turnover [338,339]. The pronounced cartilage degradation due to the inhibitory effect of ACCP on collagen IIA formation was assumed [339]. However, according to some data, cartilage degradation can be caused not only by the presence of ACCP, but also by high RA activity [5,339].

Regardless of the presence or absence of ACCP, the study of synovial biopsy specimens reveals a variety of synovitis variants. Histologic, cellular, and gene expression analyses revealed the following synovial ‘pathotypes’—lymphoid with diffuse or follicular lymphoid infiltrates and germinal centers, myeloid with less abundant lymphoid aggregates, and presence of macrophages, pauci-immune (‘low inflammatory’) pathotype with minimal infiltrating immune cells, and fibroid variants with hyperplastic processes in the absence of lymphoid aggregates and with minimal lymphoid infiltration [294,340]. It is noteworthy that the drivers of fibroid pathotype—highly activated fibroblast-like synoviocytes—are involved in pannus formation and cartilage and bone destruction [341]. Gene expression analysis in the RA joint tissues of patients undergoing total joint replacement surgery allowed van der Pouw Kraan et al. [342] to identify the following vastly different pathotypes—first group with a high inflammatory gene expression signature, including those indicating the specific activity of B and T cells, genes of cytokine receptors, cytokine/stat-activation pathway factors, HLA class II-encoding genes, and other IFNg-induced genes, a number of genes that are required for lymph node development. Upon further analysis, this group split into subgroup A, with high expression of immunoglobulin genes and genes involved in the adaptive immune response, and another one (B) with the signs of activation of the classical pathway of complement activation and high expression of genes involved in the production and degradation of extracellular matrix components. The second group included the samples with a lower expression of genes involved in inflammation and complement activation but with higher expression of genes indicative for fibroblast dedifferentiation, as well as collagen genes, with type II and XI collagen most exclusively in the RA. In the samples of both the second group and first (B) subgroup, tissues expressed genes that are involved in tissue remodeling. The association of the samples with ACCP-positive or -negative RA phenotypes was not analyzed.

## 7. OA in Full Swing

Despite the initial fundamentally different origins of RA and OA, a significant number of studies have demonstrated data suggesting possible shared mechanisms in the pathogenesis of these diseases [343,344].

In the discordance of data, caused in particular by differences in approaches to the formation of cohorts and the selection of research methods, some probable patterns emerge.

**First,** the well-known RA hallmark of citrullinated peptides and ACCP turned out to be not strictly specific. The citrullinated peptides (epitopes) of proteoglycan aggrecan were found in normal articular cartilage and were the target for ACCP [345]. In early OA patients, plasma citrullinated protein levels detected by spectrometric methods was five times higher than those in healthy controls and even higher than those in early RA (eRA) serum (4-fold increase), and unlike eRA patients, persons with early OA were ACCP-negative in this cohort [346]. However, it seems that as OA progresses, autoABs of post-translationally modified proteins are revealed, though less often than in RA. Xie and coauthors (2021) obtained serum ACCP in 5.9% of advanced OA patients (70%—in advanced (a)RA); antibodies against carbamylated peptides (aCarp), in as many as 18% of patients (47.8—in aRA); RF in 5.9% in the OA group (63.1—in aRA); and in synovial fluid, ACCP in 45.9% (71.4 in RA) and aCarp in 1.9% in OA (23.8% in RA), both cohorts consisting of patients with reliable diagnoses [347]. Unfortunately, there are very few publications on this issue, and we failed to find publications with results comparing the frequency of ACCP antibodies in the early and advanced stages of OA within the same study. We can assume several options for the relationship between OA and ACCP: (1) these antibodies are exclusively sanitary with the function of removing modified proteins; (2) involvement of adaptive immunity in the progression of OA leads to transformation into an autoimmune response; (3) and, less likely, despite the compliance of the cases included in the study with the diagnostic criteria for OA, the presence of ACCP is due to a combination with mild and unrecognized RA.

**Second,** both innate and adaptive immune systems were found to be involved in OA pathogenesis. It seems that, as the disease progresses, the predominance of innate immune responses is replaced by adaptive ones, although both are present at all stages of OA.

The current model of OA development assumes the following logic of events—local damage to the articular cartilage leads to appearance of danger-associated molecular patterns (DAMPs), due to macrophage activation and production of proinflammatory cytokine and chemokines, which in turn attract a new recruitment of macrophages as well as lymphocytes to the joint. A rampage of proinflammatory cytokines and chemokines also leads to an increase in angiogenesis, chondrocyte dysregulation, and release of metalloproteinases. In a vicious cycle of these events, the release of more proinflammatory cytokines and prostaglandins induces more cartilage destruction [348,349]. Inflammatory cell infiltrate with CD68 and CD4+ cells, neoangiogenesis, and expression of proinflammatory cytokines were demonstrated in synovial tissue samples of patients with knee pain, normal radiographs, and cartilage destruction revealed at arthroscopy at the very-early OA stage, and were significantly less pronounced in late OA synovial samples [336]. In another study, when comparing synovial biopsies obtained from patients with different OA experiences, it was found that, in the samples of early OA patients (knee pain for at least one year, no Xray OA), there were signs of thickening of the lining layer, proliferation of the lining cells, and mononuclear infiltration principally with macrophages present especially in areas of chondral defects, while in late OA samples, the thickening of the lining layer and proliferation of lining cells were not necessarily localized to areas of chondral defects and tended to be diffuse, and lymphoid cell infiltrates near newly formed vessels instead of macrophage infiltration was found [304]. Proinflammatory cytokines TNF alpha, IL1 beta, and IL6 were found in both early and late OA synovia using various methods, though there are discrepancies in the results of comparing the expression of these cytokines at an early and late stage of the disease [350,351]. Increased levels of IL-6 and TNF alpha and their association with cartilage loss were demonstrated in OA sera [352,353]. According to Barker et al. (2014), TNF-α levels were higher in early OA compared to late OA patients [354].

In the study of both early and late OA synovial samples, attributes of activation of the innate mechanisms of inflammation were found—the expression of pattern recognition receptors, matrix metalloproteinases (MMP), and Nod-like receptor pyrin domain 3 (NLRP3) inflammasomes, creating vicious cycles with pro-inflammatory cytokines [355,356,357,358,359,360,361]. It is noteworthy that, for example, when comparing RA and OA synovial samples, no differences were found in the expression of TLR 1 and 4 and adapters (domain-containing adapter protein MyD88 adapter-like, and TIR domain-containing adapter-inducing interferon/TIR-containing adapter molecule-1 adapters) predominantly in myeloid and plasmacytoid dendritic cells, and, to a lesser extent, in CD68+ type A lining cells/macrophages [356]. Noteworthily, when looking for matches of RA and OA pathogenesis, OA-associated gene SNPs of some innate immunity factors were revealed—of TNF α-308G/A gene related to excessive TNF α levels in synovia and individual susceptibility to and severity of early-onset knee OA in Egyptian females [362,363]; several SNPs of IL1 gene clusters, associated with erosive interphalangeal as well as with severe hip and knee OA [364,365,366]; several SNPs of IL6 gene, associated with radiographic hand osteoarthritis [367]; SNPs of several MMP genes associated with knee OA risk [368]; SNPs of various PRRs related with increased OA risk [61,369,370,371,372]; and excessive TLR expression in OA articular chondrocytes [60].

Other innate immunity players presented in both early and late OA synovia in significant numbers are CD8+ natural killers (nearly 30% of the CD45+ mononuclear cell infiltrate in late OA synovia) [373]. Yet the functional status of these cells was demonstrated to be different in early and late stages. In early OA samples, the cells with pro-inflammatory IFN-γ and IL-17A-producing phenotype dominated, while in late OA samples, a significant increase in CD8+IL4+ was revealed, with the phenotype consistent with postactivation exhaustion [373,374]. Other researchers revealed that the expression of perforin and granzyme B in late OA cells (CD56brightCD16(−) cells) was low and correlated with a poor cytotoxic potential against K562-sensitive target cells [375]. This is the difference between NKs in OA and highly aggressive cells in much greater numbers infiltrating RA synovia [376,377].

Lymphoid nodular aggregates are a sign that brings together synovial lesions in OA and RA. They seem to be more characteristic of late RA, in which they were found in 30% of samples (up to 65% of samples with severe disease) [378,379], while in early OA synovial specimens, they were rare or absent [350]. According to some researchers, the histological features of the most severe inflamed late OA synovial samples were identical to those found in the synovia of RA patients [378]. In particular, lymphocytic nodular aggregates, distributed around blood vessels (as in RA) according to some features, were similar to those in RA synovia [380].

Krenn and coauthors (1999) described an inflammatory infiltration of OA synovia as aggregates of cells either as small perivascular lymphoid clusters with plasma cells surrounding the lymphocytes or as small groups of plasma cells, located in the vicinity of small blood vessels [381]. Immunohistochemical analysis demonstrated CD20+ B and CD4+ and CD8+ T lymphocytes in the cluster center, surrounded by IgG (predominantly) or IgA and IgM plasma cells. The absence of proliferating Ki-67-positive cells and follicular dendritic cells in these clusters indicated that they were not ectopic germinal centers—a hallmark of RA synovia [283]. The absence of germinal centers in OA synovia was confirmed by several researchers [378].

On the other hand, full-fledged ectopic germinal centers were found not in all, but in 10–23% of RA synovia samples with a high inflammatory grade [326,382,383]. Analysis of OA B-cell V(H) gene repertoire sequence demonstrated a high number of somatic mutations and high ratios of replacement to silent mutations of synovial B lymphocytes. Additionally, V(H) gene repertoire was mismatched in synovial and blood B cells. Based on these results, Krenn and coauthors assumed that the cells underwent germinal center reactions at different sites [381]. So, according to the authors, OA memory B cells underwent germinal center reaction at different sites and migrated into the synovial tissue with subsequent differentiation into plasma cells but without further V gene diversification (accumulative type—probably a single one typical for OA). Unlike OA, the V(H) gene repertoire sequence analysis of RA synovial germinal centers and blood B cells demonstrated both accumulative and maturative (immigration of naïve B lymphocytes into synovial tissue, a local proliferation and somatic mutation of V genes in germinal centers) patterns of B-cell activation [283]. Apparently, RA is characterized by local production of autoantibodies in the synovium, in particular, of RF and antibodies to a number of organ-specific antigens; as for OA, this issue needs further investigation [283].

The most abundant cell population in OA synovial infiltrates (after macrophages—65% of the immune cells) are T lymphocytes—up 22% of the infiltrate [379]. It should be noted that T-cell infiltration was found not only in late OA synovial samples, but in early OA ones as well [384]. CD3+ cells were found to be CD4+ helpers, expressing memory/activity markers—CD28; or complex CD45RO, HLA-DR, and CD69; or complex CD69, CD25, HLA class II, CD38, CD43, and CD45RO [48,376,380]; and remarkable Th1 marker expression (CXCR3+, CCR5+) [385] and proinflammatory cytokines and their receptors [384,386]. Synovial CCR5+ and CCR3+ CD3+CD4+CD8-Th1 cell infiltration was associated with knee OA severity [385].

The comparison of the profiles of T-lymphocyte late OA and RA of the synovium led a number of researchers to the conclusion that the difference between CD4 T cells infiltrating synovia was not qualitative, but a quantitative aspect [48] with a similar distribution in synovia, but a smaller number of infiltrating CD4+ cells in OA [386].

In particular, CD+3 T cells in late OA lymphocytic nodular aggregates were stained for the activation antigens (CD69, CD25, HLA class II, CD38, CD43, and CD45RO); according to some researchers, they were localized around blood vessels as in RA, and in some instances, were indistinguishable from those found in RA, though in OA, the number of these cells and expression of activation molecules were expressed to a lesser extent than in RA synovia [380].

Most of the T cells in OA synovial aggregates are T-helper/inducer cells, whereas cytotoxic/suppressor T cells located in the periphery were found sparsely, unlike RA, in which these cells are in abundance and extremely aggressive [387]. Nevertheless, both OA and RA blood T-cell responses were induced by cartilage proteoglycan aggrecan epitopes—amino acid regions 16–39 and 263–282 [388]. Autologous chondrocytes as a whole were also shown to serve as targets for OA blood T lymphocytes, and this effect was partially blocked by antibodies against HLA class I, class II, CD4, or CD8 [389].

Another subpopulation of T cells presented both in OA and RA synovial infiltrate are Treg. The results of testing their number and activation profiles are contradictory, as well as the role of these cells in OA pathogenesis, which might be due to the cohort peculiarities [48,385,390]. Some publications point out that only minor differences were found in this subpopulation in RA and OA synovial infiltrates [391].

It is curious that few and conflicting data indicated the possible involvement of follicular helper T cells, inducing B cells to produce immunoglobulins in lymph nodes in OA pathogenesis [392]. Increased numbers of these cells and their production of IL21 in blood and synovia were found to be due to RA progression [393,394,395]. In a single publication, it was specified that Tfh cells ((CD4(+)CXCR5(+)ICOS(+)T cells)) found in RA synovia samples were absent in those of OA patients [393]. However, in this publication, a single synovia sample of a patient with unknown OA parameters and two RA samples were studied. At that time, Shan et al. (2017) demonstrated increased numbers of CXCR5+CD4+ cells, PD 1+CXCR5+CD4+, ICOS+CXCR5+CD4+, and IL 21+CXCR5+CD4+ blood T cells and increased serum IL-21 levels in 40 newly diagnosed OA patients (Kellgren and Lawrence grades II–IV) compared to healthy controls [396].

**Third.** It is well known that RA synovia has some features of a local neoplastic process, such as significantly expressed angiogenesis and aggressive expansion of dedifferentiated fibroblasts with abnormal overexpression of embryonic genes (wnt5A as an example [330]). The leading role in pannus formation is played by fibroblasts with the participation of synovial macrophages. Fibroblasts are effectors of cartilage and bone destruction due to unique invasive abilities, with production of large amounts of matrix-degrading enzymes and resistance to apoptosis. RA is characterized by an increase in the amount of antiapoptotic proteins, Bcl-2 and Mcl-1, SUMO1, FLIP, as well as by somatic mutation of the p53 protein gene, which probably contributes to the inhibition of synovicyte apoptosis [331,332,333,397].

However, pannuses did not appear to be a unique RA feature; in particular, they are a common finding in OA articular tissues, though with a lesser degree of fibroblast hyperplasia, fibrosis, and lymphocyte infiltration and a significantly lower rate of plasma cell infiltration and a milder vascular proliferation than that in RA pannus [398]. The authors of the cited publication stated pannus in OA synovia under microscopic examination was indistinguishable from that in RA.

Shibakawa et al. [399] revealed pannus-like tissue of vascular fibrous types in 90% of 15 late OA knee and hip synovia. Immunostaining revealed a predominant number of IL-1beta and MMP3 and few CD68-positive cells. The same pannus types, and additionally, with similar qualitative metabolic characteristics and pro-inflammatory cytokine response (IL-1beta, IL-8, IL-10, IL-12, TNF-alpha, IFN-gamma determined in supernatants of tissue cultures and COMP, type II collagen, TNF-alpha, IL-10, and Ki-67 expression detected by immunohistochemistry), were described by Furuzawa-Carballeda et al. in synovial samples of late OA and RA obtained during arthroplasty [400]. With a pronounced similarity of pannus in these diseases, there was a difference that was quite natural from the point of view of their pathogenesis—OA cartilage, synovial tissue, and pannus had a lower production of proteoglycans, type II collagen, and IL-1beta. The hallmark of RA pannus—neovascularization studied by testing of vascular endothelial growth factor Ets 1—was less expressed in OA samples vs. RA ones [401]. Another endothelial growth factor—VEGF—was also expressed both in RA and to a lesser extent in OA synovial samples [402]. Another important sign of pannus expression of antiapoptotic and prooncogenic factors demonstrated both in RA and OA synovial samples—in particular, oncofetal glycosylated Fn in correlation with hyperplasia [403]; Ki-67 both in RA (26.6-fold vs. histologically normal synovium) and OA (3.9-fold) pannuses [404]; and metastasis-associated protein S100A4 in RA and OA synovial tissues, in contrast with normal synovium [405].

Studies of OA synovial biopsies reveal synovial lining hyperplasia, macrophage and lymphocytic infiltration, neoangiogenesis, and fibrosis detected in varying degrees in early and late OA, and often entirely indistinguishable from those in RA [16,406]. Given the initial mechanisms of OA development, it is not surprising that synovitis in OA occurred primarily in association with the zones of cartilage and bone lesion being due to the activation of proinflammatory factors and matrix metalloproteinases by detritus [406,407]. Oehler et al. identified the following synovial variants in early and late disease stages: hyperplastic (with synovial lining and villous hyperplasia as a single process), fibrotic (with activated fibroblasts as the major players), detritus-rich (with fragments of cartilage and bone), and inflammatory (diffuse or perivascular aggregated lymphocyte and plasma cell infiltration independent on detritus presence) [408].

OA, like RA, is a multifactorial disease and also probably has different phenotypes due to the peculiarities of pathogenesis with a greater or lesser contribution of genetic factors, with the peculiarities of maturation of chondrocytes in the perinatal period, with the influence of certain non-genetic factors (trauma, obesity, etc.). This potentially leaves an imprint on the histological features of OA synovia. In particular synovial lymphoid aggregates were demonstrated to be more typical for patients with a “mechanical” than for post-traumatic OA [16]. Also, inflammatory-like OA is close to a true inflammatory arthritis and OA with cartilage remodeling features [409].

## 8. The Efficiency of Conventional Synthetic and Biologic DMARDs in RA and OA

The analysis revealed a convergence of several of pathogenic mechanisms in RA and OA, including inflammation as a key driver of both diseases. It is natural to compare the efficiency of the widely used and largely effective RA therapy with conventional or biological disease-modifying antirheumatic drugs in these two diseases. The widely used methotrexate for RA and anti-inflammatory cytokine antibodies along were tried and analyzed for OA therapy [410].

Methotrexate and anti-inflammatory cytokine antibodies, widely used in RA and tried for OA therapy, were analyzed.

Erosive hand OA (EHOA) seemed to be an obvious target for MTX/Biologic therapy due to the striking similarities with RA—a greater contribution of inflammation to erosion progression compared to other OA phenotypes, pain syndrome, and functional disability comparable to that in RA. The results of few histological studies of biopsy material demonstrated pathological changes in the synovium indistinguishable from those in RA [411,412,413,414,415].

Transferring ideas from RA therapy to OA application, despite seemingly obvious premises, led to disappointing results. EULAR experts concluded that “Patients with hand OA should not be treated with conventional or biological disease-modifying antirheumatic drugs” (level of agreement 8.8 out of 10, grade of recommendation “A”) [416]. Without disputing such a professional conclusion, we tried to understand the reasons for the failures.

An analysis of publications on this issue revealed the following (Table 3).

Methotrexate or biologic DMARD therapy was carried out in cohorts of individuals resistant to standard OA therapy. In the most studies, the indicator of drug effectiveness was the impact on pain syndrome and functional disability in comparison with common OA therapy. Most studies were conducted in advanced OA cohorts.

Patients with severe pain syndrome and advanced stages of joint damage were selected for such an aggressive therapy. The reasons for the experimental immunosuppressive and anti-inflammatory DMARD therapy failure might be the following. In addition to the inflammatory process, well-known causes of pain and functional disability in advanced OA might be destruction of articular cartilage with exposure of subchondral bone and decreased ability of articular surfaces to withstand loads. This leads to increased pressure on the bone, bending of bone beams towards the spongy bone, and their microfractures. Of certain importance are fibrosis of the capsule with compression of nerve endings and the reaction of ligaments and tendons (secondary periarthritis), which manifests itself when they are stretched during movements in the affected joint [417]. So, it would be unlikely to expect any impact from the therapy being tested. Additionally, in such cases, a significant, perhaps the major, contribution to the pain syndrome is made by sensitization of the nervous system and cognitive mechanisms. Some authors distinguish such cases into a separate OA phenotype of “chronic pain” [418]. The argument in favor of this understanding of the situation is the efficiency of therapy with anti-NGF (nerve growth factor) antibodies [419,420]. It should be noted that similar mechanisms—sensitization of the nervous system and degeneration of articular tissues—were indicated as one of the causes of refractory RA as well [421].

Another endpoint evaluated in some clinical trials was the therapy impact on the number of swollen joints and/or ultrasound or MRI signs of synovitis (effusion, synovial hypertrophy) and progression of eroding in affected joints. Contrary to expectations, conflicting data were obtained on the impact of the therapy on these indicators.

Continuing the comparison of the two analyzed diseases, it should be noted that the efficiency of conventional or biological disease-modifying antirheumatic drugs was evaluated by their ability to suppress the inflammatory process. Undoubtedly, the contribution of powerful inflammation to the pain syndrome and functional insufficiency in RA is enormous compared to low-grade inflammation in OA. Undoubtedly, the contribution of powerful inflammation to the pain syndrome and functional disability in RA is enormous compared to low-grade inflammation in OA. Moreover, unlike OA, this is a systemic, pronounced process, the monitoring of which is carried out using a well-developed system of clinical and laboratory indicators. However, when the tools used in clinical trials to assess the anti-inflammatory effect in OA cohorts were applied to assess the effectiveness of RA therapy, it turned out that, in patients with clinical remission or significant reduction in RA activity under infliximab therapy, MRI signs of low-grade inflammation in the affected joints persisted [422,423].

A significant proportion of RA patients are non-responders to conventional or biological DMARD drugs. Predictors of non-responsiveness were RA duration, erosive process in the joints, and poor functional status. In a cohort of patients with 10 years of RA experience, the percentage of non-responders/poor responders to MTX was as much as 42.6% [424]. Moreover, the same dependence of the effectiveness of MTX therapy on the duration of symptoms in months was found in the eRA cohort [425]. In early seronegative RA cohort, the starting of conventional synthetic DMARDs within 3 months after the first joint swelling led to good/moderate EULAR response in 66% patients, while erosions and poor functional status were predictors of non-responsiveness [426].

The term “Difficult-to treat RA patient” (D2T) is used in the cases of non-responsiveness to biological therapy, being the second line after failed MTX therapy. As much as 40% of RA patients did not respond to biologic therapy [421]. The predictors of D2T appeared to be the same—inadequately late start of MTX therapy, impaired physical function, and presence of erosions at baseline [427,428,429,430,431].

So, if the characteristic of non-responsiveness patterns in RA can be applied to OA, patients included in clinical trials had all chances of becoming non-responders to MTX and biologic therapy (Table 3). The exception is the cohort with inclusion of OA patients with a less pronounced erosive process in the joints (67% patients with knee OA with the Kellgren and Lawrence grade II). And incomplete but reliable effect of MTX on ultrasound signs of joint inflammation was demonstrated [432].

So, it might make sense to continue research into the efficacy of conventional or biological DMARD drugs in OA, by thoroughly rethinking the inclusion criteria for clinical trial cohorts and, what is an even more difficult problem, by selecting adequate tools to assess the effect of therapy.

**Table 3 ijms-26-08742-t003:** Application of conventional synthetic (methotrexate) and biologic DMARDs to the treatment of OA patients.

OA Phenotype	Efficacy
Methotrexate	
symptomatic, radiographic (Kellgren–Lawrence grades 3 to 4), painful inadequate response to current medication Knee OA, *n* = 207, 6 months, up to 25 mg [433]	⇓ pain, MS, ⇑ function
Knee OA, *n* = 160, 6 months, up to 25 mg radiograph (X-ray) tibiofemoral OA within the last 2 years, Kellgren–Lawrence grades 3 to 4 [11]	⇓ pain
Knee OA, *n* = 58, 4 months, 7.5 mg DS OA 2 years, Kellgren–Lawrence 2–3, synovitis [434]	no effect on pain, no difference in paracetamol consumption
Knee OA with insufficient pain relief from, or inability to tolerate, traditional analgesics including NSAIDs and opioids with synovitis, average duration 4 year, K/L score 1–4 (*n* = 20 II score 67%, *n* = 3 III score—10%), *n* = 24 weeks, up to 20 mg Erosions, 51.6; osteophytes, 68.9% [432]	13/30 (43%) achieved ≥30% reduction in pain VAS, 7 (23%) achieved ≥50% reduction, and 4 (13%) had worsened. All had synovitis (effusion or synovial hypertrophy 52 mm) at baseline and 25/30 demonstrated both pathologies. US at the final study visit (including three participants who withdrew after 12 weeks) demonstrated synovitis in 22 people. There was a median (IQR) reduction in total synovial thickness of 1.3 mm (0.7 to 3.8) (*n* = 26) and a median (IQR) reduction in total effusion measurement of 0.6 mm (1.3 to 3.6) (*n* = 26) (*p* > 0.05). Baseline synovitis or effusion (whether total values summated across the three knee compartments or maximum individual compartment scores) were not substantively correlated with baseline pain or change in 48-h pain VAS at 24 weeks (r < 0.2). Changes in synovitis and effusion at 24 weeks were similarly not substantively correlated with changes in pain.
Moderate to severe knee OA, Kellgren–Lawrence score of III to IV, *n* = 100, 6 months, 7.5 mg up to 15 mg [435]	reduced pain severity and improved functional status and quality of life
Clinical and radiographic knee OA, *n* = 155, 50% K-L grade 3–4, 12 months, 10 mg up to 25 mg [436]	⇓ pain (contradictory results) and stiffness, ⇑ function No change in synovial volume (MRI)
Knee OA with pain resistant to paracetamol, Kellgren–Lawrence II–III, *n* = 58, experience 2 years, 4 months, 7.5 mg [434]	no amelioration of symptoms functional status, tendency to reduce consumption of analgesics
Knee OA with effusion-synovitis, *n* = 215, Kellgren–Lawrence score II-22 (21%), III-39 (37%), IV-44 (42%), 52 weeks, up to 15 mg [437]	VAS pain and effusion-synovitis and maximal area, cartilage defects—no difference with placebo
Knee and hand OA, *n* = 465, 6 months, metanalysis [438]	reduced knee and hand stiffness at the end of follow-up knee and hand stiffness at 6 months of follow-up
Erosive hand OA, *n* = 64, 6 and 12 months, 10 mg [10] Verbruggen–Veys anatomical score [439] and Ghent University Score System (GUSS) scores [440]	Comparable effect of MTX and placebo on pain, functional disability, joint damage progression vs. placebo Joints with space loss appeared to be eroding less in the MTX group compared to the placebo group. Only serum IL-6 level and presence of synovitis at inclusion (but not pain, sex, age adipokines) were associated with a higher risk of erosive evolution in the non-erosive joints using the GUSS score at 12 months in the entire population.
Hand OA, *n* = 202, Kellgren and Lawrence grade ≥ 2 with synovitis, experience 6 years, 6 months, 20 mg [441]	moderate effect on reducing pain, but not function
Hand OA refractory to usual treatments [10]	
Erosive hand OA, 2 months, 10 mg [11]	decreased pain and morning stiffness, but not functional indices, number of tender and swollen joints
**Biologics**	
**Tocilizumab anti-IL6 receptor**	
Symptomatic hand OA with synovitis, Kellgren–Lawrence grade ≥ 2, experience 9 years, *n* = 104 [9]	no more effective than placebo for pain relief, number of painful and swollen joints, duration of morning stiffness, patients’ and physicians’ global assessment and function scores
Hip, knee, and hand OA (late in the most analyzed studies) Meta-analysis [13]	**Anti-TNFa**, *n* = 427, experience 6–14 (Anakinra, Adalimumab, Etanercept, Infliximab) no effect on pain and function
	**Anti-IL-1**, *n* = 404, experience (when specified) 5–11 years (AMG108, Canakinumab, ABT981, Lutikizumab) no effect on pain and function
	**Anti-NGF (nerve growth factor)**, *n* = 1749, experience 3–7 (Tanezumab, Fulranumab, Fasinumab, AMG403) ⇓ pain; ⇑ function
**Anti-IL-1**	
Knee OA, *n* = 1240, Kellgren–Lawrence grades II–III (50/50%, when specified) meta-analysis [442]	superior to placebo in terms of pain relief and functional improvement (ABT981, AMG 108, Orthokine, ABT-981, Anakinra, Canakinumab, Diacerein)
Diacerein Knee OA *n* = 1277 meta-analysis [443]	pain and function—short-term residual effectiveness
Diacerein Knee OA *n* = 1732 meta-analysis [444]	⇓ pain
Diacerein Knee OA *n* = 1533 [445]	⇑ function
Diacerein knee and/or hip meta-analysis [446]	⇓ pain, ⇑ function, ⇑ escape medication use
**TNF inhibitors**	
**Adalimumab**	
Hand OA, *n* = 276, erosive inflammatory phenotype Meta-analysis (Etanercept, Adalimumab), *n* = 276 [447]	no effect on pain at 4–6 weeks and 24–26 weeks and on grip strength at 12 months reducing progression of structural outcomes (X-ray, ultrasonography, or MRI) in hand OA with of inflammation but not in those without inflammation at 12 months
hand OA refractory to analgesics, *n* = 85, 13 years [448] Kellgren–Lawrence grade and Verbruggen–Veys anatomical scores—progression was not analyzed in dynamic	no difference to placebo for pain decrease in the number of swollen joints adalimumab group
erosive hand OA with synovitis, *n* = 43, MRI-detected synovitis [449]	No effect pain, function, and stiffness subscales from baseline to 4, 8 and 12 weeks, no effect on MRI-detected synovitis and bone marrow lesions pain and inflammation are not responsive to TNF α inhibition
erosive hand OA (on radiology), *n* = 60, experience > 6 years [450] Verbruggen–Veys anatomical scores Exploration of potential risk factors for more erosive disease—disease duration, palpable effusion at baseline	Effect on progression of joint damage in joints with soft tissue swelling compared to placebo. Risk factors for progression were then identified and the presence of palpable soft tissue swelling at baseline was recognized as the strongest predictor for erosive progression. In this subpopulation at risk, statistically significant less erosive evolution on the radiological image (3.7%) was seen in the adalimumab treated group compared to the placebo group
**Etanercept**	
symptomatic erosive inflammatory hand osteoarthritis, *n* = 90, experience 8 years [451] Verbruggen–Veys score and MRI	did not relieve pain effectively after 24 weeks in erosive osteoarthritis. Small subgroup analyses showed a signal for effects on subchondral bone in actively inflamed joints, but future studies to confirm this are warranted less MRI bone marrow lesions in more pronounced inflammatory joint group
erosive (≥ 1 IPJ with radiographic pre(erosive) anatomical phase (“J”/“E”) according to Verbruggen–Veys system) inflammatory (≥1 IPJ with soft swelling/erythema and with positive power Doppler at US) symptomatic (VAS pain > 30/100 on NSAID use, flare after NSAID washout) OA were included [452] quantitative Ghent University Scoring System	No effect-VAS pain, hand function (FIHOA), quality of life (SF-36), no. of tender joints and grip strength, radiographic progression after 4, 8, 12, 24, 36 weeks, and 1 year Symptomatic and inflammatory patients completing the study ETN was superior over placebo both on pain and structural damage assessed by GUSS; ETN was especially effective in joints with signs of inflammation

⇓—decrease in indicator; ⇑—increase in indicator.

## 9. Synthesis of the Key Findings on RA and OA Convergence and Divergence of the Pathogenetic Mechanisms

We analyzed three initially different pathologies developing in joint tissues: ACCP-positive RA, classical ‘antigen-driven’ pathology, developing in synovia with no signs of inflammatory process; ACCP-negative RA, starting with synovial inflammation triggered by nonspecific factors that becomes a chronic process due to inherited innate immune factor peculiarities; and OA, starting with inadequate chondrocyte functioning and cartilage degradation with inflammation as a driving force (Figure 5).

However, notable coincidences in RA and OA development were revealed:Shared mutations of 29 genes, encoding molecules involved in immunoinflammatory processes and ECM production.Unidirectional association of non-genetic factors with OA and ACCP-negative RA; signaling pathway overactivation with the same consequences for RA and OA.Serum ACCPs were rarely detected in OA (ACCP-negative RA exists as well!).For a clearer understanding, studies of OA variants with potentially different mechanisms are needed. Erosive hand OA is especially interesting.Innate and adaptive immune responses (although less aggressive than in RA) are involved in OA development.Identical to those in RA, lymphoid nodular aggregates (but not GCs) were revealed in 30% of OA synovial samples. On the other hand, GCs were not revealed in all RA synovial ‘pathotypes’, but only in lymphoid ones, while myeloid and especially pauci-immune and fibroid pathotypes look quite acceptable for OA.Indistinguishable from that in RA, pannuses were found in OA articular tissues.The identified list of coincidences may be evidence of evolution of some variants of OA in RA, especially from the point of view of some researchers, RA may be a syndrome developed as a result of a number of different diseases [453,454].

## Figures and Tables

**Figure 1 ijms-26-08742-f001:**
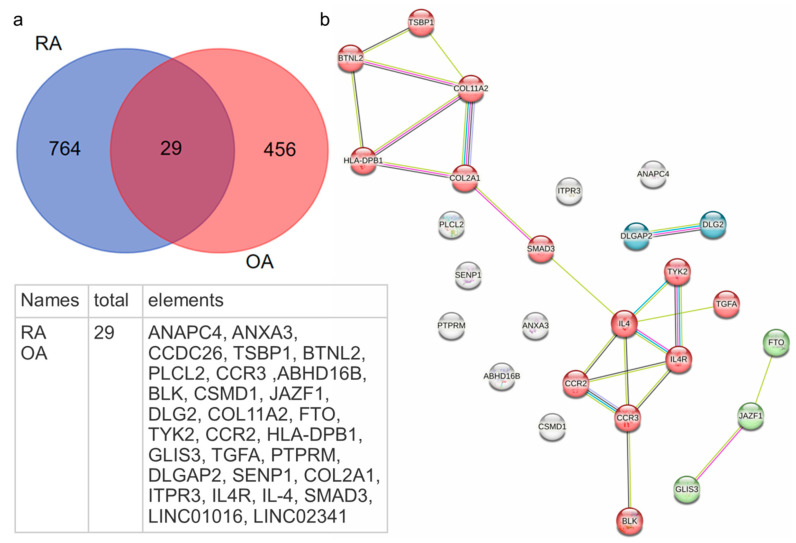
Coincidences of genetic background in OA and RA. (**a**) Shared mutated genes (GWAS). (**b**) Clusters of shared genes (STRING). Rheumatoid arthritis (RA); osteoarthritis (OA); Genome-wide association studies (GWASs); Search Tool for the Retrieval of Interacting Genes/Proteins (STRING); anaphase-promoting complex subunit 4 (ANAPC4); annexin A3 (ANXA3); CCDC26 long non-coding RNA (CCDC26); testis-expressed basic protein 1 (TSBP1); butyrophilin-like 2 (BTNL2); phospholipase C-like 2 (PLCL2); C-C motif chemokine receptor 3 (CCR3); abhydrolase domain-containing 16B (ABHD16B); BLK proto-oncogene, Src family tyrosine kinase (BLK); CUB and Sushi Multiple Domains 1 (CSMD1); JAZF zinc finger 1 (JAZF1); discs large MAGUK scaffold protein 2 (DLG2); collagen type XI (COL11A2); FTO alpha-ketoglutarate-dependent dioxygenase (FTO); tyrosine kinase 2 (TYK2); C-C motif chemokine receptor 2 (CCR2); major histocompatibility complex, class II, DPbeta 1 (HLA-DPB1); GLIS family zinc finger 3, (GLIS3); transforming growth factor alpha (TGFA); protein tyrosine phosphatase receptor type M (PTPRM); transforming growth factor alpha; DLG-associated protein 2 (DLGAP2); SUMO-specific peptidase 1 (SENP1); type II collagen (COL2A1); inositol 1,4,5-trisphosphate receptor type 3 (ITPR3); interleukin 4 receptor (IL4R); interleukin 4 (IL-4); SMAD family member 3 (SMAD3); long intergenic non-protein coding RNA 1016 (LINC01016); long intergenic non-protein coding RNA 2341 (LINC02341).

**Figure 2 ijms-26-08742-f002:**
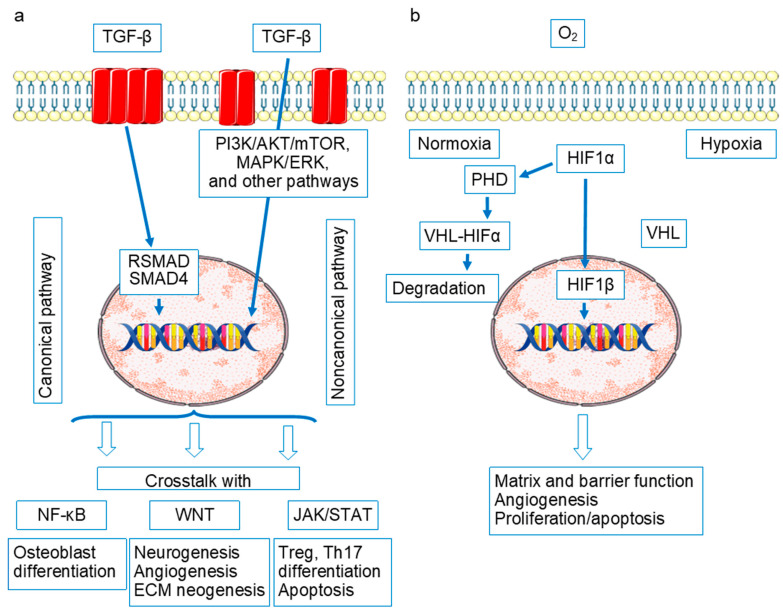
Schematic representation of the activation of signaling pathways and the influence on the pathogenesis of OA and RA. (**a**) Canonical and non-canonical activation of TGFβ signaling pathways. In canonical activation, 2 types of receptors form a hetero-tetrameric complex. Type I cell surface receptor phosphorylates intracellular receptor-regulated SMADS (R-SMADS), such as SMAD2 or SMAD3. The phosphorylated R-SMADS then binds to SMAD4. RSMAD/SMAD complexes act as transcription factors. Non-canonical, nonSMAD pathways are activated without formation of a hetero-tetrameric complex. These nonSMAD pathways include various branches of the ERK/MAPK and PI3K/AKT/mTOR pathways. (**b**) Transcriptional activity of HIF1 depends on the oxygen level. Under normoxia, the VHL-mediated ubiquitin protease pathway degrades HIF1A. Enzymes prolyl hydroxylase (PHD) and HIF prolyl hydroxylase (HPH) are involved in specific post-translational modification of HIF1A proline residues, which allow for VHL association with HIF1A. Under hypoxia, HIF1A protein degradation is prevented, and HIF1A associates with HIF1B to exert transcriptional effects on target genes. Transforming growth factor (TGF); phosphatidylinositol-3-kinase (PI3K); SMAD family member (SMAD); receptor-regulated SMADs (RSMADs); extracellular signal-regulated kinases (ERK); mitogen-activated protein kinases (MAPK); mammalian target of rapamycin (mTOR); protein kinase B (AKT); nuclear factor Κb (NF-κB); Wnt family member (Wnt); Janus kinase (JAK); signal transducer and activator of transcription (STAT); hypoxia-inducible factor 1 (HIF-1); prolyl hydroxylase (PHD); von Hippel–Lindau tumor suppressor (VHL); extracellular matrix (ECM). The information about signaling pathways was obtained from Kyoto Encyclopedia of Genes and Genomes (KEGG).

**Figure 3 ijms-26-08742-f003:**
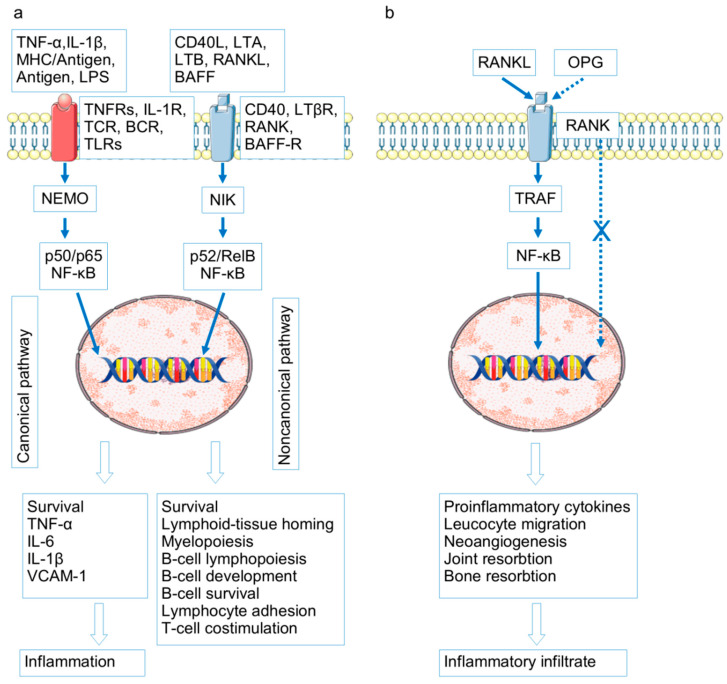
Schematic representation of the activation of signaling pathways and their influence on the pathogenesis of OA and RA. (**a**) Canonical and non-canonical activation of NF-κB signaling pathway. In canonical NEMO-dependent activation, the dimer regulating transcription—p50/p65. In non-canonical NIK-dependent activation, the dimer—p52/RelB. (**b**) Activation of the RANKL/RANK/OPG pathway. RANKL binds to RANK as its receptor and eventually leads to osteoclast precursor maturation. OPG is known as a decoy receptor for RANKL that prevents RANKL-RANK binding and the subsequent reactions. Osteoarthritis (OA); rheumatoid arthritis (RA); nuclear factor κB (NF-κB); NF-κB essential modulator (NEMO); NF-κB-inducing kinase (NIK); receptor activator for nuclear factor κB ligand (RANKL); receptor activator for nuclear factor κB (RANK); osteoprotegerin (OPG); tumor necrosis factor (TNF); TNF receptor-associated factor (TRAF); major histocompatibility complex (MHC); interleukin 1 beta (IL-1b); lipopolysaccharide (LPS); cluster of differentiation 40/cluster of differentiation 40 ligand (CD40/CD40L); lymphotoxins alpha and beta (LTA/LTB); lymphotoxin beta receptor (LTBR); B-cell activating factor receptor (BAFFR); B-cell activating factor (BAFF); interleukin 6 (IL-6); vascular cell adhesion molecule 1 (VCAM-1). The information about signaling pathways was obtained from Kyoto Encyclopedia of Genes and Genomes (KEGG).

**Figure 4 ijms-26-08742-f004:**
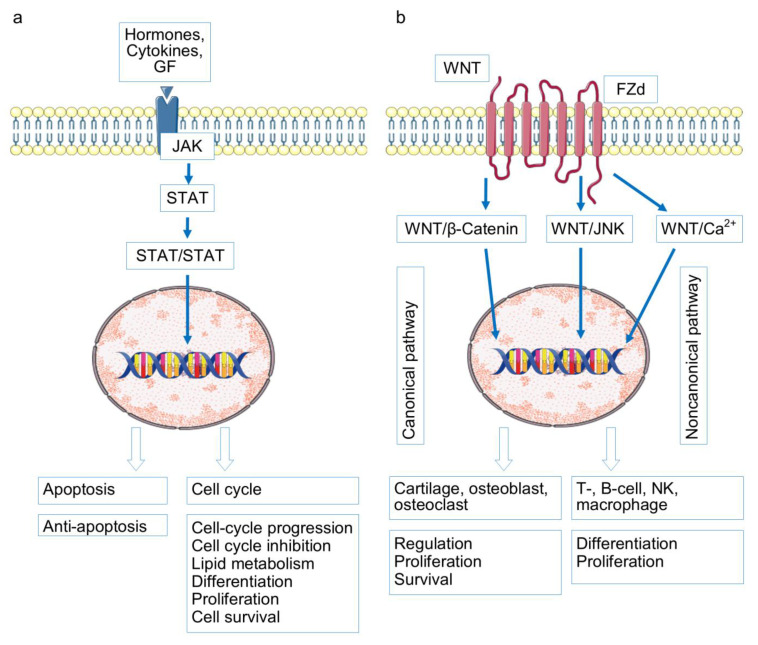
Schematic representation of the activation of signaling pathways and their influence on the pathogenesis of OA and RA. (**a**) Activation of the JAK/STAT pathway. JAK/STAT signaling involves three main proteins: cell surface receptors, JAKs, and STATs. After a ligand binds to a receptor, JAKs phosphorylate the receptor. This attracts STAT proteins, which are also phosphorylated and bind to form a dimer. The dimer translocates into the nucleus, binds to DNA, and induces gene transcription. (**b**) Activation of Wnt signaling pathways: the canonical Wnt pathway, the non-canonical planar cell polarity pathway, and the non-canonical Wnt/calcium pathway. All pathways are activated by binding of a Wnt protein ligand to a receptor of the Frizzled family. The canonical pathway involves the protein beta-catenin, while the non-canonical pathways act independently. Janus kinase (JAK); signal transducer and activator of transcription (STAT); growth factor (GF); Wnt family member (Wnt); Frizzled family receptor (FZd); c-Jun N-terminal kinases (JNK); natural killer (NK). The information about signaling pathways was obtained from Kyoto Encyclopedia of Genes and Genomes (KEGG).

**Figure 5 ijms-26-08742-f005:**
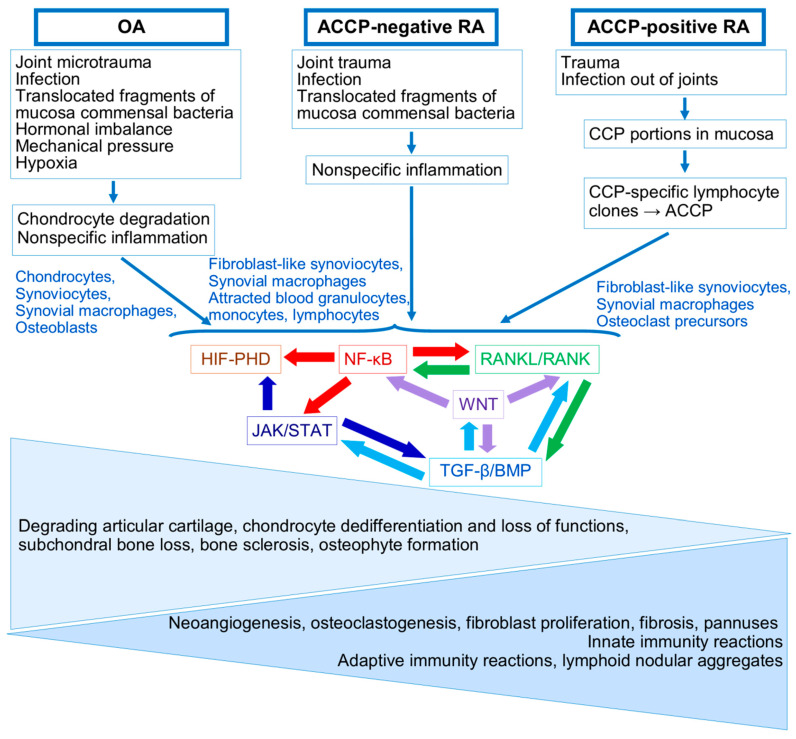
Main nodes of OA and ACCP-positive and ACCP-negative RA pathogenesis: similarities and differences. OA and ACCP-negative RA: nonspecific factors trigger inflammation and degradation of joint tissues. ACCP-positive RA: CCPs provoke ACCP production and CCP-specific lymphocyte clones in MALT, infiltrating healthy joints and generating chronic inflammation and joint tissue degradation. At early stages, a tangle of overactivated interdependent signaling pathways is the same in OA and RA joint cells; however, the lists of major players differ. In advanced OA and RA stages, signs of damage to joint tissues exhibit remarkable similarities yet differ in the intensity of certain processes. Osteoarthritis (OA); rheumatoid arthritis (RA); cyclic citrullinated peptides (CCPs); anti-CCP antibodies (ACCP); mucosa-associated lymphoid tissue (MALT). The information about interdependence of signaling pathways was obtained from Kyoto Encyclopedia of Genes and Genomes (KEGG).

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
