# Peer review of "Joint Tissues: Convergence and Divergence of the Pathogenetic Mechanisms of Rheumatoid Arthritis and Osteoarthritis"

_ijms, 2025, doi:10.3390/ijms26178742_

Round 1

Reviewer 1 Report (New Reviewer)

Comments and Suggestions for Authors

This is a traditional and comprehensive analytical review that discusses the convergence and divergence of the pathogenetic mechanisms of rheumatoid arthritis (RA) and osteoarthritis (OA) by integrating genetic factors, individual and environmental factors with the processes of disease development in joint tissues. It further analyzes the treatment of OA patients based on methotrexate and biologic DMARDs, which is of great significance for future research. There are still some questions or suggestions regarding this article:

  1. In Figure 1, the table in the lower left corner is numbered. The terms “Genome-wide association studies (GWAS); Search Tool for the Retrieval of Interacting Genes/Proteins (STRING)” appear in the figure. Please confirm whether their inclusion is necessary and whether the legend is accurately used.

In Table 1, there are some special symbols such as “?,⇑” It is recommended to provide annotations at the end of the table.

Finding the pathogenetic mechanisms of rheumatoid arthritis and osteoarthritis is a question of interest and a priority in joint research. This is a hot topic in research and the paper is well written,discusses the convergence and divergence of the pathogenetic mechanisms of rheumatoid arthritis (RA) and osteoarthritis (OA) by integrating genetic factors, individual and environmental factors with the processes of disease development in joint tissues. Nonetheless, I would like the authors to consider the following points in views to complete their study and improve the manuscript to gain clinical relevance:

  1. The manuscript provides a comprehensive overview of the pathogenetic mechanisms underlying rheumatoid arthritis (RA) and osteoarthritis (OA), and the proposed "convergence and divergence" framework is notably original. Nevertheless, the paper lacks overarching summarizing conclusions. Specifically, does the analysis of genetic similarities/differences and the subsequent signaling-pathway resemblance imply that RA and OA share a genetic basis for convergence, which later diverges due to individual susceptibility and environmental factors?
  2. Many protein-protein interactions (PPIs) retrieved from STRING (line 11) remain experimentally unverified. These data alone are insufficient as primary supporting evidence. Please supplement with citations from literature that experimentally and functionally validate the key interactions discussed.
  3. This section primarily discusses alterations from an OA perspective, with limited parallel analysis of pathways converging with or diverging from RA. Furthermore, the statement that "cellular and humoral immune responses contribute to a greater extent to RA pathogenesis" raises the important question: to what extent are such immune responses involved in OA pathogenesis? Clarifying this distinction is crucial for the convergence/divergence thesis.
  4. The flow between sections can be abrupt. For instance, at line 940, the manuscript abruptly shifts to "Lessons from the application of conventional synthetic (methotrexate) and biologic DMARDs to the treatment of OA patients." If this is intended as a distinct subsection, please clarify its purpose within the manuscript's structure and ensure a smoother thematic transition from the preceding content.
  5. Likewise, the “Individual and environmental factors” subsection lists influential factors (sex, diet, smoking, obesity, etc.), it lacks synthesis. Please address: Is there evidence for interaction between these factors ?What is the specific link between adipokines (highlighted as important in OA pathogenesis) and the listed individual/environmental factors ?
  6. In Figure 1, the table in the lower left corner is numbered. The terms “Genome-wide association studies (GWAS); Search Tool for the Retrieval of Interacting Genes/Proteins (STRING)” appear in the figure. Please confirm whether their inclusion is necessary and whether the legend is accurately used.
  7. Table 1: Special symbols (e.g., "?", "⇑") are used. Please add explanatory footnotes directly below the table defining each symbol.
  8. Istrongly recommend including a dedicated Discussion/Conclusion section at the end. This section should:Synthesize the key findings on convergence and divergenceï¼›Address current challenges and limitations in RA and OA therapyï¼›Outline promising future strategies for prevention and treatment (e.g., targeting specific inflammatory mediators, pathways identified in the convergence/divergence analysis)ï¼›Discuss the potential role of modifiable factors, such as dietary interventions (e.g., impact of omega-3 fatty acids, antioxidants), in mitigating joint tissue damage in both diseases.

Author Response

Dear Reviewer

This is a traditional and comprehensive analytical review that discusses the convergence and divergence of the pathogenetic mechanisms of rheumatoid arthritis (RA) and osteoarthritis (OA) by integrating genetic factors, individual and environmental factors with the processes of disease development in joint tissues. It further analyzes the treatment of OA patients based on methotrexate and biologic DMARDs, which is of great significance for future research. There are still some questions or suggestions regarding this article:

  1. In Figure 1, the table in the lower left corner is numbered. The terms “Genome-wide association studies (GWAS); Search Tool for the Retrieval of Interacting Genes/Proteins (STRING)” appear in the figure. Please confirm whether their inclusion is necessary and whether the legend is accurately used.

Confirmed

In Table 1, there are some special symbols such as “?,⇑” It is recommended to provide annotations at the end of the table.

Annotation added

Ý - under the influence of the factor, the adipokine level decreased

ß  –  under the influence of the factor, the adipokine level increased

?– no data

Finding the pathogenetic mechanisms of rheumatoid arthritis and osteoarthritis is a question of interest and a priority in joint research. This is a hot topic in research and the paper is well written,discusses the convergence and divergence of the pathogenetic mechanisms of rheumatoid arthritis (RA) and osteoarthritis (OA) by integrating genetic factors, individual and environmental factors with the processes of disease development in joint tissues. Nonetheless, I would like the authors to consider the following points in views to complete their study and improve the manuscript to gain clinical relevance:

  1. The manuscript provides a comprehensive overview of the pathogenetic mechanisms underlying rheumatoid arthritis (RA) and osteoarthritis (OA), and the proposed "convergence and divergence" framework is notably original. Nevertheless, the paper lacks overarching summarizing conclusions. Specifically, does the analysis of genetic similarities/differences and the subsequent signaling-pathway resemblance imply that RA and OA share a genetic basis for convergence, which later diverges due to individual susceptibility and environmental factors?

The section “Discussion/Conclusion” was structured to better summarize the results. The results of analysis are also presented graphically (Figure 5).

  1. Many protein-protein interactions (PPIs) retrieved from STRING (line 11) remain experimentally unverified. These data alone are insufficient as primary supporting evidence. Please supplement with citations from literature that experimentally and functionally validate the key interactions discussed.

Fixed this information based on publications found in a broader search.

Involved in both pathologies, long non-coding RNAs – LINC02341 and LINC01016 (lncRNAs), are involved in regulation of processes of in cell survival, apoptosis and lipid metabolism [45,46, Doi: 10.18413/2658-6533-2021-7-3-0-2].

  1. This section primarily discusses alterations from an OA perspective, with limited parallel analysis of pathways converging with or diverging from RA. Furthermore, the statement that "cellular and humoral immune responses contribute to a greater extent to RA pathogenesis" raises the important question: to what extent are such immune responses involved in OA pathogenesis? Clarifying this distinction is crucial for the convergence/divergence thesis.

The signs of immune system involvement in comparison with those in RA are discussed in the section 7. OA in full swing

Added a detailed discussion of the thesis "cellular and humoral immune responses contribute to a greater extent to RA pathogenesis":

The signaling pathways being RA hallmark and the application points of its modern biologic and targeted therapy are the drivers OA as well. Overactivation of these signaling pathways is associated with the involvement of inflammation and immune processes in the pathogenesis of both diseases (indications of immune system involvement in the pathogenesis of OA are discussed below). Yet there is an obvious difference. RA is an immune-inflammatory disease characterized by a powerful systemic inflammatory process (Goldblatt F, O’Neill SG. Clinical aspects of autoimmune rheumatic diseases. Lancet. 2013;382:797-808. doi: 10.1016/S0140- 6736(13)61499-3, Wang L, Wang FS, Gershwin ME. Human autoimmune diseases: a comprehensive update. J Intern Med. 2015;278:369-95. doi: 10.1111/joim.12395). OA is characterized by low-grade inflammation, which is not always determined outside the joint tissues and has less pronounced signs of innate and cellular immune mechanisms involvement – (DOI: 10.1038/nrrheum.2016.136).

  1. The flow between sections can be abrupt. For instance, at line 940, the manuscript abruptly shifts to "Lessons from the application of conventional synthetic (methotrexate) and biologic DMARDs to the treatment of OA patients." If this is intended as a distinct subsection, please clarify its purpose within the manuscript's structure and ensure a smoother thematic transition from the preceding content.

Changed:

The efficiency of conventional synthetic (methotrexate) and biologic DMARDs in RA and OA

The analysis revealed a convergence of a number of pathogenic mechanisms in RA and OA, including inflammation as a key driver of both diseases. Though it’s natural to compare the efficiency of the widely used and largely effective RA therapy with conventional or biological disease-modifying antirheumatic drugs in these two diseases.

  1. Likewise, the “Individual and environmental factors” subsection lists influential factors (sex, diet, smoking, obesity, etc.), it lacks synthesis. Please address: Is there evidence for interaction between these factors? What is the specific link between adipokines (highlighted as important in OA pathogenesis) and the listed individual/environmental factors?

The interplay of triggers and protectors in the pathogenesis of OA is extremely interesting and extensive. We have previously analyzed this problem for RA (Interplay of Environmental, Individual and Genetic Factors in Rheumatoid Arthritis Provocation. doi: 10.3390/ijms23158140). Unfortunately, data on the role of triggers in the pathogenesis of OA are less numerous and, moreover, contradictory. Discussion of this problem is possible within the framework of a separate extensive review. Currently, we are launching our own large project on OA research, within the framework of which we intend, in particular, to write reviews on this topic, as well as on the topic of OA treatment (including omega-3 fatty acids, antioxidants and other therapy).

Including an extensive discussion of such controversial issues in the submitted manuscript would, in our opinion, obscure the stated purpose of the publication.

The research publications on the problem of interplay of adipokines and trigger factors in OA and RA pathogenesis are few (mostly reviews) and also contradictory. Table 1 Adipokines role in RA and OA presents more or less unambiguous results obtained by different authors. The problem of the relationship between adipokines and such a trigger factor as birth weight is also undoubtedly of interest. Having tried to include an analysis of this problem in the review, we realized that these data should be considered in relation to maternal adipokines and other intrauterine problems within the framework of a separate review.
In Figure 1, the table in the lower left corner is numbered. The terms “Genome-wide association studies (GWAS); Search Tool for the Retrieval of Interacting Genes/Proteins (STRING)” appear in the figure. Please confirm whether their inclusion is necessary and whether the legend is accurately used.

Table 1: Special symbols (e.g., "?", "⇑") are used. Please add explanatory footnotes directly below the table defining each symbol. Added

Ý - under the influence of the factor, the adipokine level decreased

ß  –  under the influence of the factor, the adipokine level increased

?– no data

  1. Istrongly recommend including a dedicated Discussion/Conclusion section at the end. This section should: Synthesize the key findings on convergence and divergenceï¼›

The section title suggested by the reviewer is incomparably better than we used. The section “Discussion/Conclusion” was structured to better summarize the results. The results of analysis are also presented graphically (Figure 5).

Address current challenges and limitations in RA and OA therapy. Outline promising future strategies for prevention and treatment (e.g., targeting specific inflammatory mediators, pathways identified in the convergence/divergence analysis)ï¼›Discuss the potential role of modifiable factors, such as dietary interventions (e.g., impact of omega-3 fatty acids, antioxidants), in mitigating joint tissue damage in both diseases.

Discussion of this problem is possible within the framework of a separate extensive review. Currently, we are launching our own large project on OA research, within the framework of which we intend, in particular, to write reviews on this topic, as well as on the topic of OA treatment (including omega-3 fatty acids, antioxidants and other therapy).

Including an extensive discussion of such controversial issues in the submitted manuscript would, in our opinion, obscure the stated purpose of the publication.

  1. Synthesize the key findings on RA and OA convergence and divergence of the pathogenetic mechanisms

We analyzed three initially different pathologies developing in joint tissues: ACCP-positive RA – classical ‘antigen-driven’ pathology, developing in synovia with no signs of inflammatory process; ACCP-negative RA, starting with synovial inflammation triggered by nonspecific factors which becomes a chronic process due to inherited innate immune factor peculiarities, and OA, starting with inadequate chondrocyte functioning and cartilage degradation with inflammation as a driving force (Figure 5).

However, notable coincidences in RA and OA development were revealed:

  1. shared mutations of 29 genes, encoding molecules involved in immunoinflammatory processes and ECM production;
  2. unidirectional association of non-genetic factors with OA and ACCP-negative RA; signaling pathway overactivation with the same consequences for RA and OA.
  3. Serum ACCPs were rarely detected in OA (ACCP-negative RA exists as well!).
  4. For a clearer understanding, studies of OA variants with potentially different mechanisms are needed. Erosive hand OA is especially interesting.
  5. Innate and adaptive immune responses (although less aggressive than in RA) are involved in OA development.
  6. Identical to that in RA, lymphoid nodular aggregates (but not GCs) were revealed in 30% of OA synovial samples. On the other hand, GCs were not revealed in all RA synovial ‘pathotypes’, but only in lymphoid ones, while myeloid, and especially pauci-immune and fibroid pathotypes look quite acceptable for OA.
  7. Indistinguishable from that in RA, pannuses were found in OA articular tissues.
  8. The identified list of coincidences may be evidence of evolution of some variants of OA in RA, especially from the point of view of some researchers, RA may be a syndrome developed as a result of a number of different diseases [450,451].

Reviewer 2 Report (New Reviewer)

Comments and Suggestions for Authors

    It is well-accepted that rheumatoid arthritis is a systemic autoimmune disease characterized by synovial hyperplasia, whereas osteoarthritis is a degenerative disease characterized by reduction of the articular cartilage. But these two joint diseases commonly develop inflammation in the joints. Although it is easily speculated that common genes are operating at the phase of joint inflammation, most studies for these two joint diseases have been focussed on disease-specific mechanisms of pathogenesis and pathophysiology. 

   This paper is challenging because the authors focus on common mechanisms. Such a rare perspective might expand therapeutic applications and improve the outcome.  

But there are several concerns;

1.  More than 400 references were collected and Tables summarizing facts reported in these reference papers are useful. But important papers(4 papers below) indicating current understanding of stepwise progress of rheumatoid arthritis are lacking. A brief description of pathophysiology of rheumatoid arthritis based on these papers will be helpful to understand the pathophysiology of osteoarthritis distinct or common with rheumatoid arthritis.     

McInnes IB, Schett G. Pathogenetic insights from the treatment of rheumatoid arthritis. Lancet 2017;389:2328-37.

McInnes IB, Schett G. The pathogenesis of rheumatoid arthritis. N Engl J Med 2011;365(23):2205-19. doi: 10.1056/NEJMra1004965

Firestein GS, McInnes IB. Immunopathogenesis of Rheumatoid Arthritis. Immunity 2017;46:183-96.

Smolen JS, Aletaha D, Barton A, et al. Rheumatoid arthritis. Nat Rev Dis Primers 2018;4:18001.

2. Throughout the text, the authors quoted the many facts reported in the reference papers, which were intended to indicate convergence and divergence of the pathogenic mechanisms. Unfortunately, they just constitute the list and the final massage from this paper is obscure in spite of those facts. There is a room to be improved by further editing. 

For example, judging from the contents of this paper, focusing on unique points such as autoimmunity(including autoantibodies, germinal centers, and lymphoid nodular aggregates), pannus formation in osteoarthritis, and therapeutic effects of methotrexate and biologics on osteoarthritis, may strengthen this paper. 

Comments on the Quality of English Language

There are many typos and lack of punctuations.  And  several sentences are difficult to understand. 

Line 369  parvous → parous

Line 372  parvois → parous

Line 375  havea → have a

Line 403  “SE” shared epitope?

Line 431  in in → in in

Line 448  CCP-negative → ACCP-negative

Line 489  vistatin → visfatin

Line 497  process ad- → process, ad-

Line 498~ The sentence is difficult to understand.

Blood and synovial fluid levels of proinflammatory adipokines and similar in functions multidirectional adiponectin, promoting cartilage degradation, were increased and correlated with RA activity and OA progression.

Line 501  decreased blood → decreased in blood

Line 505~The sentence is difficult to understand.

In OA increased apelin levels were demonstrated and joint damage promotion via the same mechanism.

Line 505 (Table) → 1 or 2 should be indicated.

Line 566  cyclic citrullinated peptide (CCP) → citrullinated peptide (CP)

Line 638 Th1 macrophages → Does this mean M1 macrophages?

Line 738,741,743 eRA, aRA, aCarp;  If “e” and “a” are abbreviations, full spells should be indicated where they first appear.  

Line 829~ The long sentence, which is difficult to understand.

Based on the V(H) gene repertoire sequence analysis of OA B-cells, demonstrating a high number of somatic mutations and high ratios of replacement to silent mutations of synovial B-lymphocytes and V(H) gene repertoire mismatch in synovial and blood B-cells Krenn and coauthors assumed that the cells have undergone germinal center reactions at different sites.

Line 847  CD+3, CD+4  →CD3+, CD4+.

Author Response

Reviewer 2

    It is well-accepted that rheumatoid arthritis is a systemic autoimmune disease characterized by synovial hyperplasia, whereas osteoarthritis is a degenerative disease characterized by reduction of the articular cartilage. But these two joint diseases commonly develop inflammation in the joints. Although it is easily speculated that common genes are operating at the phase of joint inflammation, most studies for these two joint diseases have been focussed on disease-specific mechanisms of pathogenesis and pathophysiology. 

   This paper is challenging because the authors focus on common mechanisms. Such a rare perspective might expand therapeutic applications and improve the outcome.  

But there are several concerns;

  1. More than 400 references were collected and Tables summarizing facts reported in these reference papers are useful. But important papers (4 papers below) indicating current understanding of stepwise progress of rheumatoid arthritis are lacking. A brief description of pathophysiology of rheumatoid arthritis based on these papers will be helpful to understand the pathophysiology of osteoarthritis distinct or common with rheumatoid arthritis.  

Thank you. The reference were added

McInnes IB, Schett G. Pathogenetic insights from the treatment of rheumatoid arthritis. Lancet 2017;389:2328-37.

McInnes IB, Schett G. The pathogenesis of rheumatoid arthritis. N Engl J Med 2011;365(23):2205-19. doi: 10.1056/NEJMra1004965

Firestein GS, McInnes IB. Immunopathogenesis of Rheumatoid Arthritis. Immunity 2017;46:183-96.

Smolen JS, Aletaha D, Barton A, et al. Rheumatoid arthritis. Nat Rev Dis Primers 2018;4:18001.

  1. Throughout the text, the authors quoted the many facts reported in the reference papers, which were intended to indicate convergence and divergence of the pathogenic mechanisms. Unfortunately, they just constitute the list and the final massage from this paper is obscure in spite of those facts. There is a room to be improved by further editing. 

For example, judging from the contents of this paper, focusing on unique points such as autoimmunity (including autoantibodies, germinal centers, and lymphoid nodular aggregates), pannus formation in osteoarthritis, and therapeutic effects of methotrexate and biologics on osteoarthritis, may strengthen this paper. 

The section “Discussion/Conclusion” was structured to better summarize the results. The results of analysis are also presented graphically (Figure 5).

Comments on the Quality of English Language

Professor Wesley Brooks (University of South Florida, USA) is a native English speaker. He contributed to the manuscript and edited it many times. Thanks for finding the errors, they are random typos in a large text.

There are many typos and lack of punctuations.  And  several sentences are difficult to understand. 

Line 369  parvous → parous Fixed

Line 372  parvois → parous

Line 375  havea → have a

Fixed

Line 403  “SE” shared epitope?

Line 431  in in → in in

Line 448  CCP-negative → ACCP-negative

 Fixed. In addition, for consistency, in some phrases of the text, ACPA has been corrected to ACCP (both versions are found in publications)

Line 489  vistatin → visfatin

Fixed

Line 497  process ad- → process, ad-

Fixed

Line 498~ The sentence is difficult to understand.

Blood and synovial fluid levels of proinflammatory adipokines and similar in functions multidirectional adiponectin, promoting cartilage degradation, were increased and correlated with RA activity and OA progression.

The phrase has been corrected  

Line 501  decreased blood → decreased in blood

Fixed

Line 505~The sentence is difficult to understand.

In OA increased apelin levels were demonstrated and joint damage promotion via the same mechanism.

The phrase has been replaced with “In OA increased apelin levels were demonstrated due to the joint damage progression via the same mechanism - neoangiogenesis promotion”.

Line 505 (Table) → 1 or 2 should be indicated.

Fixed

Line 566  cyclic citrullinated peptide (CCP) → citrullinated peptide (CP)

Fixed

Line 638 Th1 macrophages → Does this mean M1 macrophages?

Fixed

Line 738,741,743 eRA, aRA, aCarp;  If “e” and “a” are abbreviations, full spells should be indicated where they first appear.  

Fixed

Line 829~ The long sentence, which is difficult to understand.

Based on the V(H) gene repertoire sequence analysis of OA B-cells, demonstrating a high number of somatic mutations and high ratios of replacement to silent mutations of synovial B-lymphocytes and V(H) gene repertoire mismatch in synovial and blood B-cells Krenn and coauthors assumed that the cells have undergone germinal center reactions at different sites.

The phrase was fixed “Analysis of OA B-cell V(H) gene repertoire sequence demonstrated a high number of somatic mutations and high ratios of replacement to silent mutations of synovial B-lymphocytes. Besides V(H) gene repertoire mismatch in synovial and blood B-cells.  Based on these results Krenn and coauthors assumed that the cells have undergone germinal center reactions at different sites [380].”

Line 847  CD+3, CD+4  →CD3+, CD4+.

Fixed

Round 2

Reviewer 2 Report (New Reviewer)

Comments and Suggestions for Authors  Section 9 provides the summary of this review, which has clarified the main message and improved this paper.    But  "Synthesis" may be better than "Synthesize", because it is in the subtitle.   There are several points that have not been corrected. In some sentences, inappropriate punctuation make it difficult to understand. Some suggestions to edit difficult sentences are indicated in "Comments on the Quality of English Language"          Comments on the Quality of English Language

Revised Line516~519(1st submitted Line498~) One suggestion of editing;

Promoting cartilage degradation, proinflammatory adipokines and multidirectional adiponectin in the blood and synovial fluids were increased and correlated with RA activity and OA progression.

Revised Line523~524(1st submitted Line505~) One suggestion of editing;

In OA, increased apelin levels were demonstrated due to the joint damage progression via the same mechanism – promotion of neo-angiogenesis.   Revised Line765(1st submitted Line743);

Usages of abbreviations are confusing because “a” indicates 2 meanings; advanced and antibodies. Since “Antibodies against carbamylated peptides” appears not so frequently in the later part, full spelling may be no problem. Abbreviation to “Anti-Carp” may be an alternative.

Revised Line859~863(1st submitted Line829~834) One suggestion of editing;

Analysis of OA B-cells V(H) gene repertoire sequence demonstrated a high number of somatic mutations and high ratios of replacement to silent mutations of synovial B-lymphocytes. Besides   V(H) gene repertoire mismatch in synovial and blood B-cells. Krenn and coauthors assumed that the cells have undergone germinal center reactions at different sites.

Line 887(revised)  CD+3 → CD3+  not corrected

Revised Line972~975 One suggestion of editing;

The analysis revealed a convergence of several pathogenic mechanisms in RA and OA, including inflammation as a key driver of both diseases. Since RA therapies with conventional or biological disease-modifying antirheumatic drugs have been widely used and largely effective, it is reasonable to compare the efficiency of these therapies in these two diseases.

Author Response

Dear reviewer, thank you for the valuable comments. A native English speaker - professor (University of South Florida) made a number of corrections in the text. Some unsuccessful phrases were replaced.

Comments and Suggestions for Authors

 Section 9 provides the summary of this review, which has clarified the main message and improved this paper.    But  "Synthesis" may be better than "Synthesize", because it is in the subtitle.  

Fixed

  1. Synthesize Synthesis the key findings on RA and OA convergence and divergence of the pathogenetic mechanisms

 There are several points that have not been corrected. In some sentences, inappropriate punctuation make it difficult to understand. Some suggestions to edit difficult sentences are indicated in "Comments on the Quality of English Language"         

Comments on the Quality of English Language

Revised Line516~519(1st submitted Line498~) One suggestion of editing;

Promoting cartilage degradation, proinflammatory adipokines and multidirectional adiponectin in the blood and synovial fluids were increased and correlated with RA activity and OA progression.

Revised Line523~524(1st submitted Line505~) One suggestion of editing;

In OA, increased apelin levels were demonstrated due to the joint damage progression via the same mechanism – promotion of neo-angiogenesis.   Revised Line765(1st submitted Line743);

Usages of abbreviations are confusing because “a” indicates 2 meanings; advanced and antibodies. Since “Antibodies against carbamylated peptides” appears not so frequently in the later part, full spelling may be no problem. Abbreviation to “Anti-Carp” may be an alternative.

The abbreviation has been replaced with the full spelling: Line 765

.Revised Line859~863(1st submitted Line829~834) One suggestion of editing;

Analysis of OA B-cells V(H) gene repertoire sequence demonstrated a high number of somatic mutations and high ratios of replacement to silent mutations of synovial B-lymphocytes. Besides   V(H) gene repertoire mismatch in synovial and blood B-cells. Krenn and coauthors assumed that the cells have undergone germinal center reactions at different sites.

Line 887(revised)  CD+3 → CD3+  not corrected

Fixed Line 888

Revised Line972~975 One suggestion of editing;

The analysis revealed a convergence of several pathogenic mechanisms in RA and OA, including inflammation as a key driver of both diseases. Since RA therapies with conventional or biological disease-modifying antirheumatic drugs have been widely used and largely effective, it is reasonable to compare the efficiency of these therapies in these two diseases.

This manuscript is a resubmission of an earlier submission. The following is a list of the peer review reports and author responses from that submission.

Round 1

Reviewer 1 Report

Comments and Suggestions for Authors

In this comprehensive review article, the authors have compared the pathogenetic mechanisms associated with the development of anti-cyclic citrullinated peptide antibody (ACCP)-positive rheumatoid arthritis (RA) with ACCP-negative RA and osteoarthritis (OA).  The authors present a detailed overview of genetic mutations associated with each type of arthritis and identify 29 genes overlapping both RA phenotypes with OA.  These data have been presented focusing on the signaling pathways involved in pathogenesis of each disease type, culminating in the excellent overview presented in Figure 5.  The authors have expanded from the genetic information to provide a detailed comparison the risk factors and distinct and shared pathological features of each disease type.  While this has been reviewed by others previously, the inclusion of the comparison is a strength of the current review when presented in the context of the genetic information.

While the review is an excellent contribution to the current literature and provides an outstanding overview, a short section addressing current therapeutic approaches, and those that may be developed based on the extensive information presented in the review would have strengthened an otherwise excellent manuscript.

Author Response

In this comprehensive review article, the authors have compared the pathogenetic mechanisms associated with the development of anti-cyclic citrullinated peptide antibody (ACCP)-positive rheumatoid arthritis (RA) with ACCP-negative RA and osteoarthritis (OA).  The authors present a detailed overview of genetic mutations associated with each type of arthritis and identify 29 genes overlapping both RA phenotypes with OA.  These data have been presented focusing on the signaling pathways involved in pathogenesis of each disease type, culminating in the excellent overview presented in Figure 5.  The authors have expanded from the genetic information to provide a detailed comparison the risk factors and distinct and shared pathological features of each disease type.  While this has been reviewed by others previously, the inclusion of the comparison is a strength of the current review when presented in the context of the genetic information.

Dear reviewer thank you for the for an excellent assessment of our work. We put the soul into it.

There is no Criticism requiring an answer, there is no paragraph in this

While the review is an excellent contribution to the current literature and provides an outstanding overview, a short section addressing current therapeutic approaches, and those that may be developed based on the extensive information presented in the review would have strengthened an otherwise excellent manuscript.

Short comments about possible new therapeutic approaches are added to the Introduction section (Page 3)

Reviewer 2 Report

Comments and Suggestions for Authors

The overlap in pathophysiology and underlying mechanism in OA and RA is an interesting topic for a review. Unfortunately, the present review has some significant issues which prevent it being suitable for publication. While the text lists a wide range of facts about OA and RA, the focus is too broad and the coverage too superficial for these to be successfully combined into a compelling argument about the overlap in underlying mechanisms between OA and RA. Sources or methodology for data presented in the Figures (particularly Figure 1 and 2) are not sufficiently explained, and many of the Figures provide only superficial information about established signaling pathways rather than interesting new insights into the overlap between OA and RA.

Comments on the Quality of English Language

The English language employed in the text unfortunately falls short of the mark in terms of clarity. Many statements throughout are confusingly worded, and some are dubious or untrue (notably “RA is due to secondary OA development”), possibly due to miswording or mistranslation.

Author Response

Dear reviewer,
Thanks for the strict, but extremely useful assessment of our work.

The overlap in pathophysiology and underlying mechanism in OA and RA is an interesting topic for a review. Unfortunately, the present review has some significant issues which prevent it being suitable for publication. While the text lists a wide range of facts about OA and RA, the focus is too broad and the coverage too superficial for these to be successfully combined into a compelling argument about the overlap in underlying mechanisms between OA and RA. Sources or methodology for data presented in the Figures (particularly Figure 1 and 2) are not sufficiently explained, and many of the Figures provide only superficial information about established signaling pathways rather than interesting new insights into the overlap between OA and RA.

Telegraphic style of presentation is largely due to the attempt to squeeze a large amount of information which is necessary to fully cover the problem, into the standard volume of a journal review.

We have tried to fix this defect.

The comments on the shared genetic were completely rewritten

  1. Comparison of RA and OA genetic backgrounds

Genome-wide association studies (GWAS) have provided significant insights into the shared genetic underpinnings of RA and OA. Several single nucleotide polymorphisms (SNPs) have been identified that are common to both diseases, suggesting overlapping genetic factors that influence susceptibility to joint diseases (Figure 1). Within the 1,249 genes identified, 29 genes were found to be common to both pathologies  (Figure 1A).

A number of these genes, – collagen type II (COL2A1), collagen type XI (COL11A2) and CUB and Sushi Multiple Domains 1(CSMD1), play crucial roles in cartilage integrity and extracellular matrix formation. Type II Collagen is the main extracellular matrix (ECM protein), while inclusion of a minor type, XI collagen, is due to some physical properties of mature ECM [16]. Interestingly, collagen XI is used in a rat model for the induction of chronic arthritis [17]. CSMD1 is in particular a transmembrane inhibitor of classical and lectin complement pathways [18] and as such can modulate the immune environment that might aggravate cartilage degradation [19]. SMAD family member 3 (SMAD3) and transforming growth factor alpha (TGFA) coded molecules are the factors of TGF signaling pathways which being overactivated both in OA and RA are involved in ECM neogenesis as well as in other pathological processes (neoangiogenesis, apoptosis, osteoblast differentiation, Figure 2).

Additionally, anaphase promoting complex subunit 4 (ANAPC4), SUMO specific peptidase 1 (SENP1), DLG associated protein 2 (DLGAP2), BLK proto-oncogene, Src family tyrosine kinase (BLK) and PTPRM coded factors are implicated in the dysregulation of apoptotic pathways, disrupting cell cycle regulation in synovial cells, causing increased apoptosis in chondrocytes contributing to cartilage loss in OA, and disturbed cell cycling in synovial fibroblasts, promoting hyperplasia in RA (Supplementary table 1). The ANXA3 coded molecule collaborates with RANK in acceleration of osteoblast differentiation [20] and contributes to cell proliferation, angiogenesis via the HIF-VEGF signaling pathway [21]. TYK2 as well as CSMD1 code factors modulating the JAK/STAT cascade (KEGG obtained data) [19,22], overactivated in both OA and RA DLG2 and ITPR3 genes code factors, interfering with calcium dependent signaling pathways regulating ion channel functioning (PRR signaling and inflammasome formation as examples) [23]. The physiological function of FTO alpha-ketoglutarate dependent dioxygenase (FTO) is not known yet (NCBI), but the other non-heme iron enzymes are involved in DNA and RNA damage by oxidative demethylation and interact with homeobox transcription factor iriquois-3 (IRX3) genes [24,25]. Given the possible association of the development of RA (at least its ACPA-positive variant) with errors in epigenetic events in the perinatal period, it is intriguing to consider that IRX3 is likely involved in a number of processes at the early stages of fetal development and in maturation of a number of cell line precursors [26–29]. Besides the more thoroughly studied function of FTO/IRX3 is the involvement in adipocyte precursor development and obesity and diabetes pathogenesis [30–32]. Meta-analysis revealed association of FTO gene SNPs with hip/ knee OA, collagen formation and extracellular matrix organization biological pathways  [33,34] . In addition it was demonstrated that FTO overexpression alleviates OA progression [35,36]. In RA FTO SNPs were associated with joint damage, due to the inflammation activity [37] . Increased FTO expression in RA synovial cells enhanced their proliferation and migration and decreased senescence and apoptosis [38].

Two chemokine gene SNPs, – C-C motif chemokine receptor 3 (CCR3) and C-C motif chemokine receptor 2 (CCR2) may be involved due to the attraction of leucocytes in inflammatory joint loci both in OA and RA, and together with SNPs of IL4 gene and its receptor as well as, discussed above, a SNP of TYK2, coding protein associated with the cytoplasmic domain of type I and type II cytokine receptors and being a part of IFN type I and type III signaling pathways (NCBI) demonstrate the involvement of immune system not only in RA but in OA as well. Additional notable coincidences are gene mutations which may be important for the development of the adoptive immune response both in OA and RA –genes of molecules involved in B-cell development and B-cell receptor signaling – phospholipase C like 2 (PLCL2), BLK proto-oncogene, Src family tyrosine kinase (BLK) and the two genes located in MHC, class II – butyrophilin like 2 (BTNL2), encoding type I transmembrane protein involved in immune surveillance, serving as a negative T-cell regulator by decreasing T-cell proliferation and cytokine release, and HLA-DPB1 (major histocompatibility complex, class II, DP beta 1 ) presenting peptides derived from extracellular proteins by B lymphocytes, dendritic cells, macrophages (NCBI) [33,39–44]. Mentioned above CSMD1 factor promotes both B-lymphocyte receptor signaling and pathways related to antigen presentation [19].

Among the genes involved in both pathologies, are long non-coding RNAs – LINC02341 and LINC01016 (lncRNAs). Data on the functions of these RNAs are poorly represented in PubMed publications. Yet, it looks like they promote cell proliferation activity and decrease apoptosis [45,46] (Pages 3-7)

Comments on the Quality of English Language

The English language employed in the text unfortunately falls short of the mark in terms of clarity. Many statements throughout are confusingly worded, and some are dubious or untrue (notably “RA is due to secondary OA development”), possibly due to miswording or mistranslation.

One of the co-authors of the manuscript is a native English speaker. He carefully edited the wording. Some of the poor wording and telegraphic style of presentation are largely due to the attempt to squeeze a large amount of information which is necessary to fully cover the problem, into the standard volume of a journal review. We have tried to correct this problem.

Reviewer 3 Report

Comments and Suggestions for Authors

The authors provide an interesting review comparing shared pathologies and mechanisms contributing to some forms of RA and OA. 

The manuscript in it's present form provides interesting information regarding these diseases, but some additional details are required in order to reach a broader audience and improve the flow of the work. These suggestions are outlined below:

Introduction: The introduction in the current form is very brief. The authors could greatly improve this section via the addition of pertinent information regarding the pathology, presentation, treatment, prevalence, and underlying causes of the two diseases.

 Comparisons of RA and OA genetic backgrounds: This section provides good detail regarding the roles of the 29 shared genes. The authors could improve this section by including additional references in support of the role of these genes in cartilage disease/health. Please include the full phrase single nucleotide polymorphisms before using SNPs abbreviation. 

4. Individual and environmental RA and OA triggers: This section contains some interesting information, but perhaps it could be restructured for easier reading? For instance, maybe include all female related triggers for both into one paragraph, diet-related in another and so on. Also, could the authors also include comments on the role of age in RA and OA as a trigger. 

5. Comparison of pathogenesis of ACCP+ and ACCP- RA and OA: This section discusses many pertinent details underlying the onset and progression of these diseases. Perhaps the authors might consider moving some of this material to the introduction and expanding on them to address the issue of such a brief introduction?

Concluding remarks: The authors make the point that there is evolutionary links between RA and OA. While these diseases do share similar gene and signaling pathways, there is also room for the shared genes to be a result of pro-inflammatory stimulation and cartilage degradation. This might suggest that the shared genes are a result of inflammation and not shared evolutionary paths. Could the authors comment on this? 

Author Response

Dear reviewer,
Thank you very much for the valuable and interesting comments.

Introduction: The introduction in the current form is very brief. The authors could greatly improve this section via the addition of pertinent information regarding the pathology, presentation, treatment, prevalence, and underlying causes of the two diseases.

We significantly expanded the Introduction section (pages 2-3).

1 Introduction

Rheumatoid arthritis (RA) and osteoarthritis (OA) are two widespread multifactorial diseases that affect joints. In recent years, the perception of OA as a purely degenerative process has changed. The inflammatory process in the arthritic joint is recognized as the more important component of the pathogenesis and even a trigger for OA [1]. The well-known players in RA pathogenesis and the targets for biological disease modifying antirheumatic drug, TNFα and IL-1β, were recognised as drivers of catabolic signaling in OA joints [2]. On the other hand, RA is one of the causes of secondary OA, and the RA inflammatory process can be the driving force of such OA development. Inspite of the well documented dependence of secondary OA symptoms on RA duration [3], changed serum levels of various cartilage turnover markers (N-terminal propeptide of collagen IIA, cross-linked C-telopeptide of collagen II, oligomeric matrix protein) in correlation with radiographic/MRI joint damage signs, signifying possible OA initiation, were demonstrated to occur early in RA patients [4,5] . Curiously, as much as 20% of 247 persons at the 3rd (arthalgia) and 4th (undifferentiated arthritis) preclinical RA stages exhibited joint symptoms that were provoked by unusual excessive joint activity, while among 461 early RA patients, there were none in which such activity was identified as a trigger of the disease [6]. The question arises whether imbalance appears in the pre-RA stage and whether it is one of the triggers of RA.

The recognition of inflammation as an OA driver has brought about the idea to apply the groups of drugs used in RA to OA therapy – corticosteroids, methotrexate, biologics [7–12]. The results of primarily short-term therapy (4 weeks – 6 months) presented in the references mentioned above and a number of other publications are contradictory. Therefore, there is a need for long-term trials.

An assumption can be made that contradictory results are due to differences in the patients included in the study. The difficulty is that OA as a multifactorial disease might have various phenotypes. The existence of different RA phenotypes is not in doubt. The two most well characterized phenotypes are the variants with or without antibodies against cyclic citrullinated peptides (ACCP) with the obvious differences of the initial pathogenic mechanisms discussed in this review.

Ideas regarding different variants of OA are still in the hypothesis-development stage. The following options may be suggested: post-traumatic OA with probable disturbances in repair processes after joint injury; mechanical variants due to the chronic microtrauma of the joint, aggravated by excess weight and other environmental and individual factors [13,14]; primary hand erosive osteoarthritis with a probable major contribution of immunological mechanisms in its pathogenesis (in particular, the frequent presence of lymphoid follicles in joint tissues, which brings it closer to RA) [15].

So, even a cursory glance allows one to see a certain similarity between OA and RA. These diseases may share not only the arena in which they unfold but also some pathogenic mechanisms. Therefore, it appears reasonable to consider the interweaving of OA and RA pathological mechanisms, which, to some extent, may lead to the convergence of their clinical features.

Accordingly, we will consider the similarities and differences of (i) genetic factors predisposing to the development of OA and RA, (ii) individual and environmental triggers of diseases, as well as (iii) the processes of disease development in joint tissues.

 Comparisons of RA and OA genetic backgrounds: This section provides good detail regarding the roles of the 29 shared genes. The authors could improve this section by including additional references in support of the role of these genes in cartilage disease/health. Please include the full phrase single nucleotide polymorphisms before using SNPs abbreviation. 

We completely rewrote this section taking into account the comments, supplementing it with the required information (unfortunately, in our opinion there were not enough publications for a full analysis) (Pages 3-7)

  1. Comparison of RA and OA genetic backgrounds

Genome-wide association studies (GWAS) have provided significant insights into the shared genetic underpinnings of RA and OA. Several single nucleotide polymorphisms (SNPs) have been identified that are common to both diseases, suggesting overlapping genetic factors that influence susceptibility to joint diseases (Figure 1). Within the 1,249 genes identified, 29 genes were found to be common to both pathologies  (Figure 1A).

A number of these genes, – collagen type II (COL2A1), collagen type XI (COL11A2) and CUB and Sushi Multiple Domains 1(CSMD1), play crucial roles in cartilage integrity and extracellular matrix formation. Type II Collagen is the main extracellular matrix (ECM protein), while inclusion of a minor type, XI collagen, is due to some physical properties of mature ECM [16]. Interestingly, collagen XI is used in a rat model for the induction of chronic arthritis [17]. CSMD1 is in particular a transmembrane inhibitor of classical and lectin complement pathways [18] and as such can modulate the immune environment that might aggravate cartilage degradation [19]. SMAD family member 3 (SMAD3) and transforming growth factor alpha (TGFA) coded molecules are the factors of TGF signaling pathways which being overactivated both in OA and RA are involved in ECM neogenesis as well as in other pathological processes (neoangiogenesis, apoptosis, osteoblast differentiation, Figure 2).

Additionally, anaphase promoting complex subunit 4 (ANAPC4), SUMO specific peptidase 1 (SENP1), DLG associated protein 2 (DLGAP2), BLK proto-oncogene, Src family tyrosine kinase (BLK) and PTPRM coded factors are implicated in the dysregulation of apoptotic pathways, disrupting cell cycle regulation in synovial cells, causing increased apoptosis in chondrocytes contributing to cartilage loss in OA, and disturbed cell cycling in synovial fibroblasts, promoting hyperplasia in RA (Supplementary table 1). The ANXA3 coded molecule collaborates with RANK in acceleration of osteoblast differentiation [20] and contributes to cell proliferation, angiogenesis via the HIF-VEGF signaling pathway [21]. TYK2 as well as CSMD1 code factors modulating the JAK/STAT cascade (KEGG obtained data) [19,22], overactivated in both OA and RA DLG2 and ITPR3 genes code factors, interfering with calcium dependent signaling pathways regulating ion channel functioning (PRR signaling and inflammasome formation as examples) [23]. The physiological function of FTO alpha-ketoglutarate dependent dioxygenase (FTO) is not known yet (NCBI), but the other non-heme iron enzymes are involved in DNA and RNA damage by oxidative demethylation and interact with homeobox transcription factor iriquois-3 (IRX3) genes [24,25]. Given the possible association of the development of RA (at least its ACPA-positive variant) with errors in epigenetic events in the perinatal period, it is intriguing to consider that IRX3 is likely involved in a number of processes at the early stages of fetal development and in maturation of a number of cell line precursors [26–29]. Besides the more thoroughly studied function of FTO/IRX3 is the involvement in adipocyte precursor development and obesity and diabetes pathogenesis [30–32]. Meta-analysis revealed association of FTO gene SNPs with hip/ knee OA, collagen formation and extracellular matrix organization biological pathways  [33,34] . In addition it was demonstrated that FTO overexpression alleviates OA progression [35,36]. In RA FTO SNPs were associated with joint damage, due to the inflammation activity [37] . Increased FTO expression in RA synovial cells enhanced their proliferation and migration and decreased senescence and apoptosis [38].

Two chemokine gene SNPs, – C-C motif chemokine receptor 3 (CCR3) and C-C motif chemokine receptor 2 (CCR2) may be involved due to the attraction of leucocytes in inflammatory joint loci both in OA and RA, and together with SNPs of IL4 gene and its receptor as well as, discussed above, a SNP of TYK2, coding protein associated with the cytoplasmic domain of type I and type II cytokine receptors and being a part of IFN type I and type III signaling pathways (NCBI) demonstrate the involvement of immune system not only in RA but in OA as well. Additional notable coincidences are gene mutations which may be important for the development of the adoptive immune response both in OA and RA –genes of molecules involved in B-cell development and B-cell receptor signaling – phospholipase C like 2 (PLCL2), BLK proto-oncogene, Src family tyrosine kinase (BLK) and the two genes located in MHC, class II – butyrophilin like 2 (BTNL2), encoding type I transmembrane protein involved in immune surveillance, serving as a negative T-cell regulator by decreasing T-cell proliferation and cytokine release, and HLA-DPB1 (major histocompatibility complex, class II, DP beta 1 ) presenting peptides derived from extracellular proteins by B lymphocytes, dendritic cells, macrophages (NCBI) [33,39–44]. Mentioned above CSMD1 factor promotes both B-lymphocyte receptor signaling and pathways related to antigen presentation [19].

Among the genes involved in both pathologies, are long non-coding RNAs – LINC02341 and LINC01016 (lncRNAs). Data on the functions of these RNAs are poorly represented in PubMed publications. Yet, it looks like they promote cell proliferation activity and decrease apoptosis [45,46]

 Individual and environmental RA and OA triggers: This section contains some interesting information, but perhaps it could be restructured for easier reading? For instance, maybe include all female related triggers for both into one paragraph, diet-related in another and so on. Also, could the authors also include comments on the role of age in RA and OA as a trigger. 

We have tried to structure the data presented in Table 3. Unfortunately, the different approaches to research methods presented in the publications did not allow us to sufficiently unify the information, even sacrificing some sources.  The comments were added to the section Individual and environmental RA and OA triggers including comments on the role of age in RA and OA as a trigger) (Pages 14-16).

So, two RA variants and OA are predominantly "female" family-aggregated age-related diseases. Yet their links with non-genetic triggers appeared to have certain differences, probably reflecting the difference in the contribution of concrete pathogenetic mechanisms to their development.

Some patterns emerge when comparing the lists of RA and OA triggers. Family clustering has a complex nature, including shared genetics; infections; microbiome; lifestyle; ecology[166]. Familial aggregation of RA and OA may be due to different triggers from this list.

The peak risk of developing both RA and OA occurs in the age range of 50-60 years. The well-known molecular mechanisms of aging are the same for OA and RA with a greater or lesser contribution to their development. Imbalance in production and inactivation of reactive oxygen species due to chondrocyte cell death and ECM degradation. [167,168] Oxidative stress is due to DNA, lipid and protein damage leading to synovial inflammation essential for the both diseases [169]. Reactive oxygen species are involved in in the process of carbamylation, which provokes the production of antibodies to carbamylated peptides, which have diagnostic and prognostic significance in RA [170,171]. Oxidative stress might trigger joint symptoms in patients at preclinic RA stages[172]. Immune system aging results in proinflammatory shift of immune reactions[173]. Tissue damage and increased proinflammatory potential of immune system predisposes for chronic-persistent inflammation.

Common sense would expect traditional lifestyle differences and gender-based occupational characteristics certainly contribute to the female/male disparities in RA and OA. Yet, it’s noteworthy that ACPA-negative RA variant appeared to be more dependent on fluctuations in estrogen levels than the ACPA-positive cases. The same trend was observed in OA. Indeed, menopause is characterized by decrease in estrogen and progesterone levels, and weakening of their anti-inflammatory effects [166]. Being parous is associated with OA and ACCP-negative RA, but not with ACCP-positive cases. Normal pregnancy is characterized by increased estrogen and progesterone levels and Th1→Th2 immune responses, and with the delivery process there is a powerful explosion of Th1 responses against the background of decreasing estrogen and progesterone levels. So, repetitive surges of a Th1 response at each delivery might ultimately trigger a persistent immune-inflammatory process, characteristic primarily of CCP-negative RA and, perhaps, to some extent, of OA. Breastfeeding is characterized by high levels of prolactin supporting a Th1 response by reduced levels of estrogen and prolactin compared to the period before pregnancy (which quickly return to normal in the absence of breastfeeding). So, this situation, if repeated, is also fraught with the possibility of provoking a chronic immune-inflammatory response. This is a very simplified speculative scheme of the possible involvement of sex hormone-associated processes in provocation of ACCP-negative RA and OA. Further studies are needed to clarify the possible gender-dependent differences between ACCP-positive and ACCP-negative disease variants.

Obesity and mental stress which can increase risk for OA and ACCP-negative RA, but not ACCP-positive RA risk, also have pro-inflammatory potential [166]. Another trigger of OA and ACCP-negative, but not ACCP-positive RA is physical activity, which may impact the diseases via repetitive joint tissue microtrauma and concomitant inflammation. The scheme of the greater importance of non-specific pro-inflammatory triggers of OA and ACPA-negative, but not ACPA-positive RA, contradicts the links between smoking, which provokes ACPA-positive RA, but possibly has a protective effect in ACPA-negative cases and OA, and alcohol consumption, which reduces the risk of developing ACPA-positive RA and provokes ACPA-negative cases and OA. The provoking effect of smoking in ACCP-positive RA was expected given its well-known association with ACCP production [131,154,155], however, its other known mechanisms are proinflammatory [166]. The protective effect of alcohol consumption on RA development is usually associated with its anti-inflammatory effect [166]. However, it is not clear why this effect occurs in ACPA-positive cases  but not in ACPA-negative RA or OA.

Despite the obvious contradictions, possibly related to the lack of data analyzing the differences in the connections of triggers with ACPA-positive and ACPA-negative RA and OA, a certain trend is visible - the lists of triggers for OA and ACPA-negative OA are similar.

Supplementary table 3. RA and OA links with non-genetic risk factors

ACCP-positive RA

ACCP-negative RA

OA

Family Aggregation

Familial risk OR=3.7* (95% CI 2.9-4.7), n=1,652

<40 years OR=6.2* (95% CI 3.5-10.9)

40-60 years OR=3.3* (95% CI 2.4-4.6)

60+ years OR=3.3 *(95% CI 2.0-5.3) [110]

Familial risk OR=2.1* (95% CI 1.5-3.1), n=873

<40 years OR=3.5* (95% CI 1.4-9.0)

40-60 years OR=2.1* (95% CI 1.2-3.6)

60+ years OR=1.8 (95% CI 1.0-3.3) [110]

Familial risk OR = 4.4* (95% CI 2.0-9.5), hip OR=3.9* (95% CI 1.8-8.4), spine OR = 2.2 (95% CI 1.0-5.1) adjusted for age, sex, and body mass index [111]; knee OR= 3.61 (95% CI 2,69-4.85) [112]

Age

Age (50-59, vs 30-39) HR=2.445* (95% CI 1.909-3.131) [116]

Z

Z

vs 25-44 years

45-49 RR=1.2 (95% CI 0.9-1.7)

50-54 RR=2.0* (95% CI 1.5-2.7)

55-59 RR=2.1* (95% CI 1.5-2.8)

60-64 RR=1.4 (95% CI 0.6-3.2) [113]

vs 25-44 years

45-49 RR=2.1* (95% CI 1.5-3.0)

50-54 RR=2.0* (95% CI 1.3-3.0)

55-59 RR=2.7* (95% CI 1.4-5.4)

60-64 RR=2.5* (95% CI 1.5-4.2)[113]

Prevalence of OA (%)

40 to < 45 13.2% (95% CI 10-16)

45 to < 50 16.5% (95% CI 14-19)

50 to < 55 21.8% (95% CI 19-24)

55 to < 60 27.4% (95% CI 25-30)

60 to < 65 32.9% (95% CI 30-36)

65 to < 70 36.2% (95% CI 33-40)

75 to < 80 38.8% (95% CI 34-44) [114]

Gender Associated Factors

Male vs female HR=0.387* (95% CI 0.340-0.441) [116]

Z

Male vs female OR=2.87 (95% CI 1.94-4.25) [112]

Incidence rate of knee OA IRR=0.55* (95% CI 0.32-0.94), hip OA IRR=0.64* (95% CI 0.48-0.86), and a non-significant reduction hand OA IRR=0.65 (95% CI 0.35-1.20)[117]

Gender-specific prevalence: 1.54*% (95% CI, 1.4-1.69) for females; 0.4*% (95% CI, 0.32-0.49) for males [115]

Age-standardized prevalence of hand OA modestly higher in women (44.2%) vs men (37.7%), age-standardized prevalence of erosive and symptomatic OA higher in women (9.9*% vs 3.3%, and 15.9*% vs 8.2%) [153]

Postmenopausal vs premenopausal women HR=1.2 (95% CI 0.9-1.6) [113]

Postmenopausal vs premenopausal women HR=2.1* (95% CI 1.4-3.0) [113]

Spontaneous/surgical menopause vs premenopausal OR=1.13* (95% CI 1.07-1.21)/ OR=1.18* (95% CI 1.08-1.28) [119]

Age at menopause vs pre-menopausal

≤44 years HR=1.2 (95% CI 0.9-1.7)

45-49 years HR=1.2 (95% CI 0.9-1.7)

≥ 50 years HR=1.2 (95% CI 0.9-1.7) [113]

Age at menopause vs pre-menopause)

≤44 years HR=2.1* (95% CI 1.4-3.1)

45-49 years HR=1.7* (95% CI 1.1-2.7)

≥ 50 years HR=2.0* (95% CI 1.2-3.1) [113]

Z

Postmenopausal hormones

current users vs never users HR= 1.3 (95% CI 0.9-1.8) no effect [113]

Postmenopausal hormones

current users vs never users HR= 1.3 (95% CI 0.9-1.8) no effect [113]

Postmenopausal hormones

current users vs never users OR=0.70* (95% CI 0.50-0.99) [122]

Estrogen plus progestogens therapy users vs never users OR=0.3* (95% CI 0.1-0.7) [121]

Estrogen plus progestogens therapy users vs never users OR=1.0 (95% CI 0.7-1.4) [121]

Estrogen plus progestogens users vs non users HR=1.092 (95% CI, 1.048-1.137) [124]

Estrogens therapy users vs never users OR=0.73* (95% CI 0.75-0.84)[119] ; HR=1.235* (95% CI, 1.148-1.329) [124];

risk of knee OA RR=1.8* (95% CI 1.2-2.6) [58]; risk of hip OA OR =5.03, (95% CI 1.70-14.84) hand OA OR=1.57 (95% CI 1.05-2.33) [125]

Parous vs nulliparous RR=0.96 (95% CI 0.89-1.04) [126]

Parous vs nulliparous R=1.031*(95% CI 1.023-1.039) [130]

Parous vs nulliparous women aged 18-44 years R=0.9 (95% CI 0.7-1.2)  [127]

Parous vs nulliparous women aged 18-44 years OR=2.1* (95% CI 1.4-3.2) [127]

2 children vs 1 RR=0.84* (95% CI 0.78-0.90)

3 children vs 1 RR=0.83* (95% CI 0.77- 0.91) [126]

1-2 children vs none R=1.49* (95% CI 1.22-1.82)

3-4 children vs none R=3.26* (95% CI 2.68-3.96) [129]

vs normal pregnancy history

Pregnancies complicated by hyperemesis RR=1.70* (95% CI 1.06-2.54), gestational hypertension RR=1.5* (95% CI 1.06-2.02), pre-eclampsia RR=1.42* (95% CI 1.08-1.84).

Risk of RA increased significantly with increasing number of pregnancies complicated by gestational hypertension or pre-eclampsia (p for trend = 0.003) [126]

Z

Breast feeding (yes/no) OR=0.2* (95% CI 0.1-0.7) [134]

Breast feeding (yes/no) OR=1.55* (95% CI 1.18-2.03)[129];

risk of knee OA OR=2.30* (95% CI 1.09-4.86)[135]

Breast feeding for 1-12 months vs 0 months OR=0.74* (95% CI 0.41-1.35) [200]; HR=0.66* (95% CI 0.46- 0.94) [143]

Z

Breast feeding for 13 months or more vs 0-6 months OR=0.77 (95% CI 0.63-0.94) [132]

Breast feeding for 13 months or more vs 0-6 months OR=0.91 (95% CI 0.65-1.27) [132]

Z

Breast feeding for >13 months vs 0 months 0.47* (95% CI 0.19-1.14) [306]

Breast feeding for 24 months vs 0 months OR= 0.5 (95% CI 0.3-0.7) [133]

Oral contraceptives users vs never users OR=1.65* (95% CI 1.06-2.57) [131]

Oral contraceptive users vs never users OR=1.19 (95% CI 0.68-2.07) [131]

Oral contraceptives users vs never users OR=0.9 (95% CI 0.6-1.4)[123] ;  OR=0.972 (95% CI 0.967-0.977) [130]

Oral contraceptives ever vs never users OR=0.84 (95% CI 0.74-0.96)[132]

Oral contraceptives ever vs never users OR=0.93 (95% CI 0.79-1.10) [132]

RA risk ever, current and past Oral contraceptives users vs never users OR=1.00 (95% CI 0.87-1.15), OR=0.93 (95% CI 0.70-1.23) and OR=0.93 (95% CI 0.78-1.12), respectively [307]

Z

Perinatal Factors

Birth weight

> 4.54 kg (vs 3.2-3.85 kg) RR=2.1 (95% CI 1.4-3.3)[138]

> 4 kg (vs 3-3,9) OR=3.6* (95% CI 1.4-9.1)

Large for gestational age (yes/no) OR=4.4* (95% CI 1.6-12)

Small for gestational age (yes/no) OR=1.0 (95% CI 0.3-2.6) [134]

Birth weight

Lower birth weight and osteophytes in hip OR=1.51* (95% CI 1.13-2.01)

clinical hand OA OR=1.396* (95% CI 1.05-1.85) [141]; HR=2.02 (95% CI 1.10-3.73) [140]

Preterm birth HR=2.53* (95% CI 1.30-4.92) [140]

Body Mass Index

BMI ≥25 <30 kg/m2 OR=0.8 (95% CI 0.7-1.0), women OR=1.0 (95% CI 0.8-1.2) [142]; OR=1.01 (95% CI 0.69-1.47) [131]

BMI ≥25 <30 kg/m2 OR=1.4* (95% CI 1.1-1.9), women OR=1.6* (95% CI 1.2-2.2) [142]; HR 2.75* (95% CI 1.39-5.46) for obese vs normal-weight [143]; OR=1.24 (95% CI 0.74-2.10) [131]

BMI ≥25 <30 kg/m2 hip OA RR=1.04 (95% CI 1.00-1.07) and joint surgery RR=1.16 (95% CI 1.11-1.22) not reliable [144];

>25 kg/m2 OR=3.29* (95% CI 2.40-4.51) [112]; knee OA OR=2.68* (95% CI 2.33-3.09); hip OA OR=1,65* (95% CI 1.46-1.87) [146]

≥30 kg/m2 OR=1.15 (0.62 to 2.13) [131]; 30-34.9 kg/m2 HR=0.583* (95% CI 0.440-0.772) [116]

≥30 kg/m2 OR=3.45* (95% CI 1.73-6.87) [131]

> 30 kg/m2 knee OA OR=2.81* (95% CI 1.32-5.96), hip OA OR=1.11 (95% CI 0.41-2.97), hand OA OR=2.59* (95% CI 1.08-6.19)[147] ; knee OA OR=7.48* (95% CI 5.45-10.27) [146]

Z

Z

Risk associated with a 5 kg/m2 increase in BMI: hand OR=1.25* (95% CI 1.06-1.49) [148] ; knee OA RR=1.25* (95% CI 1.17-1.35) [145]

Coffee consumption

Dose dependent effect*

0-5 cups/day 1.20 (95% CI 0.76-1.91)

5-10 cups/day 1.70 (95% CI 0.95-3.05) >10 cups/day 2.18 (95% CI 1.07-4.42) [131]

Dose dependent effect

0-5 cups/day 0.79 (95% CI 0.41-1.52)

5-10 cups/day 0.94 (95% CI 0.47-1.90)

>10 cups/day 1.23 (95% CI 0.48-3.16) [131]

1 cup/day knee OA OR=1.023* (95% CI 1.009-1.038) [149]; OR=1.01 (95% CI 0.999-1.025) [150]

Men: < 2 cup/day OR=1.13 (95% CI 0.50-2.55), 2-3 cup/day OR=1.79 (95% CI 0.81-3.97), 4-6 cup/day OR=2.21 (95% CI 0.91- 5.35), and ≥ 7 cup/day OR=3.81 (95% CI 1.46-12.45) .

Women - no association with studied doses [150]

Alcohol

Drinkers vs non-drinkers OR=0.52* (95% CI 0.36-0.76) [308]

Drinkers vs non-drinkers OR= 0.74 (95% CI 0.53-1.05) not reliable [308]

Z

Unit/day vs non-drinker HR=0.86* (95% CI 0.74-0.99) [143]

Z

5-10 cups/week vs non-drinker HR=0.810* (95% CI 0.657-0.999) [116]

Z

≥4 doses/ week vs non-drinkers OR=0.64* (95% CI 0.36-1.14) [151]

Z

0 drinks/week vs 0-5 drinks/week OR=1.18* (95% CI 0.82-1.71)

> 15 drinks/week vs > 0-5 drinks/week

OR=0.66* (95% CI 0.24-1.84) [131]

0 drinks/week vs > 0-5 drinks/week OR=0.98 (95% CI 0.62-1.55)

>15 drinks per week vs >0 to 5 drinks per week OR=1.36* (95% CI 0.54-4.66) [131]

1-6 drinks/week vs 0 drinks/week hand OA OR= 1.82* (95% CI 0.99-3.36) [152]

1-3 drinks/week vs 0 drinks/week hand OA OR= 1.55* (95% CI 0.43-2.67) [153]

Smoking

Ever smokers vs never smokers:

RR=1.5* (95% CI 0.8-2.9) - 21* (95% CI 11.0-40.2) (no SE-double SE) [154];

OR=4.1* (95% CI 1.9-9.2) [155]

Ever smokers vs never smokers:

RR= 0.6* (95% CI 0.4-1.0) - 0.8 (95% CI 0.4-1.7) (no SE-double SE)  [154];

OR=0.7 (95% CI 0.3-2.0) [155]

Ever smokers vs never smokers knee OA

RR=0.80* (95% CI 0.73-0.88) [157]

Current smoking vs nonsmokers

OR=2.13* (95% CI 1.54-2.95) [131]

Current smoking vs nonsmokers OR=1.01 (95% CI 0.65-1.57) [131]

Z

>10 to 20 pack-years OR=2.41* (95% CI 1.51-3.82) [131]

1-19 pack-years OR=3.3* (95% CI 1.1-9.8),

≥20 pack-years OR=5.5* (95% CI 1.6-17.6) [155]

>10 to 20 pack-years OR=0.72*(95% CI 0.37-1.40) [131]

1-19 pack-years OR=1.9 (95% CI 0.5-8.0),

≥20 pack-years OR=0.4 (95% CI 0.1-2.3) [155]

1-10/day knee OA OR=0.91 (95% CI 0.68-1.21) [156]

10-20/day knee OA OR=0.566* (95% CI 0.470-0.683) [158]

11-20/day knee OA OR=0.70* (95% CI 0.52-0.94) [156]

Mental stress

Depression prior RA onset (yes/no) HR=1.38* (95% CI 1.31-1.46), antidepressant use before RA onset (yes/no) HR=0.74* (95% CI 0.71-0.76) [160]

exposure during 5-year period preceding the diagnosis of RA vs no OR=8.88* (95% CI 1.6-47.6) [159]

Z

Female

Depression prior RA onset (yes/no)

HR=1.12 (95% CI 0.93-1.35) [161]

Female

Depression prior RA onset (yes/no)

HR=1.63*(95% CI 1.27-2.09) [161]

Female

Depressive mood (yes/no)

OR=2.80* (95% CI 1.31-3.31),

Psychological distress (yes/no)

OR=1.92* (95% CI 1.21-3.05)

Male

Depressive mood (yes/no)

OR=1.51* (95% CI 1.16-1.95)

Psychological distress (yes/no)

OR=1.36* (95% CI 1.07-1.72) [163]

Knee/hip

high stress perception

male OR=1.59*(95% CI 1.10-2.31)

female OR=1.41* (95% CI 1.19-1.68)

depression

male OR=1.52* (95% CI 1.01-2.29)

female OR=1.27* (95% CI 1.04-1.54) [162]

Physical activity

Physical activity at work (ten years prior to interview)

Slight - OR=1.10 (95% CI 0.65-1.87)

High - OR=1.18 (95% CI 0.68-2.06) [131]

Physical activity at work (ten years prior to interview)

Slight - OR=1.41* (95% CI 0.74-2.67)

High - OR=1.33* (95% CI 0.66-2.67) [131]

Stair climbing >10 flights/d OR=6.08* (95% CI 4.16-8.89) [112]

Walking >2 miles/day OR=1.9* (95% CI 1.4-2.8) [165]

Occupational lifting and hip OA

regular lifting of

25 kg OR=3.6* (95% CI 1.3-9.7)[164] ; OR=1.7* (95% CI 1.2-2.6) [165];

50 kg in main job OR = 4.0* (95% CI 1.1-14.2) [164]

Comparison of pathogenesis of ACCP+ and ACCP- RA and OA: This section discusses many pertinent details underlying the onset and progression of these diseases. Perhaps the authors might consider moving some of this material to the introduction and expanding on them to address the issue of such a brief introduction?

We have significantly revised the Introduction, including adding some information about ACPA-positive and ACPA-negative RA variants, as well as suggestions about possible OA variants. However, we believe it is more appropriate to discuss in more detail in separate paragraphs the initial mechanisms of development of RA variants, followed by a discussion of the OA initial stage.

Concluding remarks: The authors make the point that there is evolutionary links between RA and OA. While these diseases do share similar gene and signaling pathways, there is also room for the shared genes to be a result of pro-inflammatory stimulation and cartilage degradation. This might suggest that the shared genes are a result of inflammation and not shared evolutionary paths. Could the authors comment on this? 

Dear reviewer,

Thank you for this extremely interesting question . The information presented in the review is gene polymorphisms. We tried to trace the effect of a mutation of a specific gene on its function in OA. Unfortunately, there were not enough publications for a full analysis, and isolated publications with the results of the expression of mutated genes in this disease.

. The information presented in the review is gene polymorphisms. We tried to trace the effect of a mutation of a specific gene on its function in OA. Unfortunately, there were not enough publications for a full analysis, and isolated publications with the results of the expression of mutated genes in this disease.
